# Latent Consistency Models: Synthesizing High-Resolution Images with Few-step Inference

## Abstract

Latent Diffusion models (LDMs) have achieved remarkable results in synthesizing high-resolution images. However, the iterative sampling process is computationally intensive and leads to slow generation. Inspired by Consistency Models (Song et al., 2023), we propose Latent Consistency Models (**LCMs**), enabling swift inference with minimal steps on any pre-trained LDMs, including Stable Diffusion (Rombach et al., 2022). Viewing the guided reverse diffusion process as solving an augmented probability flow ODE (PF-ODE), LCMs are designed to directly predict the solution of such ODE in latent space, mitigating the need for numerous iterations and allowing rapid, high-fidelity sampling. Efficiently distilled from pre-trained classifier-free guided diffusion models, a high-quality $768\times768$ $2\sim4$-step LCM takes only 32 A100 GPU hours for training. Furthermore, we introduce Latent Consistency Fine-tuning (LCF), a novel method that is tailored for fine-tuning LCMs on customized image datasets. Evaluation on the LAION-5B-Aesthetics dataset demonstrates that LCMs achieve state-of-the-art text-to-image generation performance with few-step inference.

## 1 Introduction

Diffusion models have emerged as powerful generative models that have gained significant attention and achieved remarkable results in various domains (Ho et al., 2020; Song et al., 2020a; Nichol & Dhariwal, 2021; Ramesh et al., 2022; Song & Ermon, 2019; Song et al., 2021). In particular, latent diffusion models (LDMs) (e.g., Stable Diffusion (Rombach et al., 2022)) have demonstrated exceptional performance, especially in high-resolution text-to-image synthesis tasks. LDMs can generate high-quality images conditioned on textual descriptions, by utilizing an iterative reverse sampling process that performs gradual denoising of samples. However, diffusion models suffer from a notable drawback: the iterative reverse sampling process leads to slow generation speed, limiting their real-time applicability. To overcome this drawback, researchers have proposed several methods to improve the sampling speed, which involves accelerating the denoising process by enhancing ODE solvers (Ho et al., 2020; Lu et al., 2022a;b), which can generate images within $10\sim20$ sampling steps. Another approach is to distill a pre-trained diffusion model into models that enable few-step inference Salimans & Ho (2022); Meng et al. (2023). In particular, Meng et al. (2023) proposed a two-stage distillation approach to improving the sampling efficiency of classifier-free guided models. Recently, Song et al. (2023) proposed consistency models as a promising alternative aimed at speeding up the generation process. By learning consistency mappings that maintain point consistency on ODE-trajectory, these models allow for single-step generation, eliminating the need for computation-intensive iterations. However, Song et al. (2023) is constrained to pixel space image generation tasks, making it unsuitable for synthesizing high-resolution images. Moreover, the applications to the conditional diffusion model and the incorporation of classifier-free guidance have not been explored, rendering their methods unsuitable for text-to-image generation synthesis.

In this paper, we introduce "Latent Consistency Models" (LCMs), a novel approach for rapid, high-resolution image generation. Building on the foundation of Latent Diffusion Models (LDMs), LCMs utilize a pre-trained autoencoder from Stable Diffusion (Rombach et al., 2022), optimizing image generation in a lower-dimensional latent space. The core innovation lies in our one-stage guided distillation method. This process efficiently transforms a pre-trained guided diffusion model into a

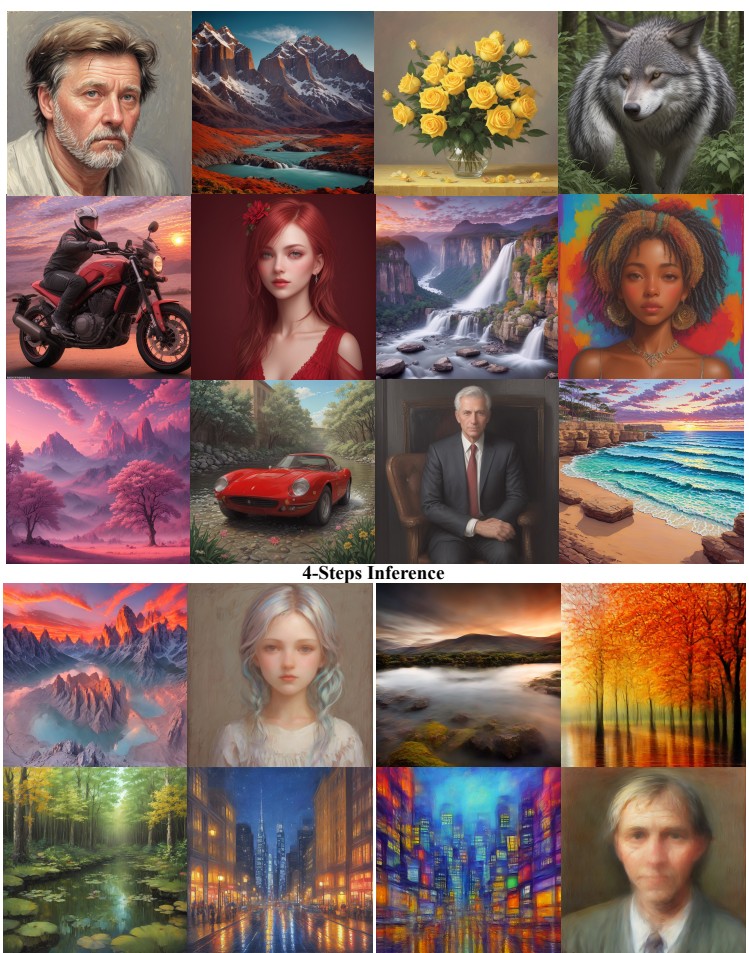

**Figure 1:** Images generated by Latent Consistency Models (LCMs). LCMs can be distilled from any pre-trained Stable Diffusion (SD) in only 4,000 training steps (∼**32 A100 GPU Hours**) for generating high quality 768×768 resolution images in 2∼4 steps or even one step, significantly accelerating text-to-image generation. We employ LCM to distill the Dreamer-V7 version of SD in just 4,000 training iterations. The corresponding prompts can be found in Appendix M.

latent consistency model. Consequently, unlike the previous approach of using iterative sampling processes based on diffusion models to solve PF-ODE, latent consistency models are tasked with directly predicting the final solution of PF-ODE. We also incorporate the technique of SKIPPING-STEP to expedite this distillation process. Additionally, we propose "Latent Consistency Fine-tuning," enabling the customization of pre-trained LCMs for specific image datasets, facilitating few-step inference with high adaptability and precision. Our main contributions are summarized as follows:

- We propose Latent Consistency Models (LCMs) for fast high-resolution image generation. LCMs employ consistency models in the image latent space, enabling fast few-step or even one-step high-fidelity sampling on pre-trained latent diffusion models (e.g., Stable Diffusion (SD)).

- We provide a simple and efficient one-stage *guided consistency distillation* method to distill SD for few-step (2∼4) or even 1-step sampling. We propose the SKIPPING-STEP technique to further accelerate the convergence. For 2- and 4-step inference, our method costs only 32 A100 GPU hours for training and achieves state-of-the-art performance on LAION-5B-Aesthetics dataset.

- We introduce a new fine-tuning method for LCMs, named Latent Consistency Fine-tuning, enabling efficient adaptation of a pre-trained LCM to customized datasets while preserving the ability of fast inference.

## 2 RELATED WORK

**Diffusion Models** have achieved remarkable success in the field of image generation, as evidenced by a series of pioneering works (Ho et al., 2020; Song et al., 2020a; Nichol & Dhariwal, 2021; Ramesh et al., 2022; Rombach et al., 2022; Song & Ermon, 2019). These models operate by being

trained to systematically remove noise from noise-corrupted data, thereby estimating the *score* of the underlying data distribution. This process, during inference, involves drawing samples through a reverse diffusion process that gradually denoises a data point. In comparison to Variational Autoencoders (VAEs) (Kingma & Welling, 2013; Sohn et al., 2015) and Generative Adversarial Networks (GANs) (Goodfellow et al., 2020), diffusion models are highly regarded for their training stability and superior capability in likelihood estimation, marking an advancement in generative models.

**Accelerating DMs.** Despite their success, diffusion models are hindered by a critical limitation: their slow generation speed. To address this bottleneck, a variety of methods have been proposed. Training-free approaches, such as ODE solvers (Song et al., 2020a; Lu et al., 2022a;b) and adaptive step size solvers (Jolicoeur-Martineau et al., 2021), along with predictor-corrector methods (Song et al., 2020b), offer some solutions. Training-based approaches like optimized discretization (Watson et al., 2021), truncated diffusion (Lyu et al., 2022; Zheng et al., 2022), neural operator (Zheng et al., 2023), and distillation (Salimans & Ho, 2022; Meng et al., 2023) have also been explored. More recent developments include new generative models designed for faster sampling (Liu et al., 2022; 2023), indicating ongoing innovation in this area.

**Latent Diffusion Models** (LDMs) (Rombach et al., 2022) have shown exceptional capabilities in synthesizing high-resolution text-to-image conversions. For example, Stable Diffusion utilizes forward and reverse diffusion processes in the data's latent space, leading to more efficient computation and higher quality outputs. The cross-attention mechanisms in LDMs, which encode text, empower these models to synthesize images that align closely with the accompanying text descriptions.

**Consistency Models** (CMs) (Song et al., 2023) represent an emerging class of generative models that offer rapid sampling capabilities while maintaining high-quality generation. CMs employ a novel consistency mapping technique that maps any point in an ODE trajectory directly back to its origin, facilitating a fast, one-step generation process. These models can be trained either by distilling pre-trained diffusion models or as standalone generative models. Compared to other one-step, non-adversarial generative models, CMs have shown superior performance on standard benchmarks, marking them as a significant contribution to the field of generative modeling. Details of CMs and their implementation are further elaborated in the subsequent sections.

## 3 PRELIMINARIES

In this section, we briefly review diffusion and consistency models and define relevant notations.

**Diffusion Models:** Diffusion models, or score-based generative models Ho et al. (2020); Song et al. (2020a) is a family of generative models that progressively inject Gaussian noises into the data, and then generate samples from noise via a reverse denoising process. In particular, diffusion models define a forward process transitioning the original data distribution $p_{data}(x)$ to marginal distribution $q_t(\boldsymbol{x}_t)$, via transition kernel: $q_{0t}(\boldsymbol{x}_t \mid \boldsymbol{x}_0) = \mathcal{N}(\boldsymbol{x}_t \mid \alpha(t)\boldsymbol{x}_0, \sigma^2(t)\boldsymbol{I})$, where $\alpha(t), \sigma(t)$ specify the noise schedule. In continuous time perspective, the forward process can be described by a stochastic differential equation (SDE) Song et al. (2020b); Lu et al. (2022a); Karras et al. (2022) for $t \in [0, T]$: $\mathrm{d}\boldsymbol{x}_t = f(t)\boldsymbol{x}_t\,\mathrm{d}t + g(t)\mathrm{d}\boldsymbol{w}_t,\ \boldsymbol{x}_0 \sim p_{data}(\boldsymbol{x}_0)$, where $\boldsymbol{w}_t$ is the standard Brownian motion, and

$$f(t) = \frac{\mathrm{d}\log\alpha(t)}{\mathrm{d}t}, \quad g^2(t) = \frac{\mathrm{d}\sigma^2(t)}{\mathrm{d}t} - 2\frac{\mathrm{d}\log\alpha(t)}{\mathrm{d}t}\sigma^2(t). \tag{1}$$

By considering the reverse time SDE (see Appendix A for more details), one can show that the marginal distribution $q_t(\boldsymbol{x})$ satisfies the following ordinary differential equation, called the *Probability Flow ODE* (PF-ODE) (Song et al., 2020b; Lu et al., 2022a):

$$\frac{\mathrm{d}\boldsymbol{x}_t}{\mathrm{d}t} = f(t)\boldsymbol{x}_t - \frac{1}{2}g^2(t)\nabla_{\boldsymbol{x}}\log q_t(\boldsymbol{x}_t),\ \boldsymbol{x}_T \sim q_T(\boldsymbol{x}_T). \tag{2}$$

In diffusion models, we train the noise prediction model $\boldsymbol{\epsilon}_\theta(\boldsymbol{x}_t, t)$ to fit $-\nabla\log q_t(\boldsymbol{x}_t)$ (called the *score function*). Approximating the score function by the noise prediction model in 21, one can obtain the following *empirical PF-ODE* for sampling:

$$\frac{\mathrm{d}\boldsymbol{x}_t}{\mathrm{d}t} = f(t)\boldsymbol{x}_t + \frac{g^2(t)}{2\sigma_t}\boldsymbol{\epsilon}_\theta(\boldsymbol{x}_t, t), \quad \boldsymbol{x}_T \sim \mathcal{N}(\boldsymbol{0}, \tilde{\sigma}^2\boldsymbol{I}). \tag{3}$$

For class-conditioned diffusion models, Classifier-Free Guidance (CFG) (Ho & Salimans, 2022) is an effective technique to significantly improve the quality of generated samples and has been widely used in several large-scale diffusion models including GLIDE Nichol et al. (2021), Stable Diffusion (Rombach et al., 2022), DALL·E 2 (Ramesh et al., 2022) and Imagen (Saharia et al., 2022). Given

a CFG scale $\omega$, the original noise prediction is replaced by a linear combination of conditional and unconditional noise prediction, i.e., $\tilde{\epsilon}_{\boldsymbol{\theta}}(\boldsymbol{z}_t, \omega, \boldsymbol{c}, t) = (1 + \omega)\epsilon_{\boldsymbol{\theta}}(\boldsymbol{z}_t, \boldsymbol{c}, t) - \omega\epsilon_{\boldsymbol{\theta}}(\boldsymbol{z}, \varnothing, t)$.

**Consistency Models:** The Consistency Model (CM) (Song et al., 2023) is a new family of generative models that enables one-step or few-step generation. The core idea of the CM is to learn the function that maps any points on a trajectory of the PF-ODE to that trajectory's origin (i.e., the solution of the PF-ODE). More formally, the consistency function is defined as $\boldsymbol{f} : (\boldsymbol{x}_t, t) \longmapsto \boldsymbol{x}_\epsilon$, where $\epsilon$ is a fixed small positive number. One important observation is that the consistency function should satisfy the *self-consistency property*:

$$\boldsymbol{f}(\boldsymbol{x}_t, t) = \boldsymbol{f}(\boldsymbol{x}_{t'}, t'), \forall t, t' \in [\epsilon, T]. \tag{4}$$

The key idea in (Song et al., 2023) for learning a consistency model $\boldsymbol{f}_{\boldsymbol{\theta}}$ is to learn a consistency function from data by effectively enforcing the self-consistency property in Eq. 4. To ensure that $\boldsymbol{f}_{\boldsymbol{\theta}}(\boldsymbol{x}, \epsilon) = \boldsymbol{x}$, the consistency model $\boldsymbol{f}_{\boldsymbol{\theta}}$ is parameterized as:

$$\boldsymbol{f}_{\boldsymbol{\theta}}(\boldsymbol{x}, t) = c_{\text{skip}}(t)\boldsymbol{x} + c_{\text{out}}(t)\boldsymbol{F}_{\boldsymbol{\theta}}(\boldsymbol{x}, t), \tag{5}$$

where $c_{\text{skip}}(t)$ and $c_{\text{out}}(t)$ are differentiable functions with $c_{\text{skip}}(\epsilon) = 1$ and $c_{\text{out}}(\epsilon) = 0$, and $\boldsymbol{F}_{\boldsymbol{\theta}}(\boldsymbol{x}, t)$ is a deep neural network. A CM can be either distilled from a pre-trained diffusion model or trained from scratch. The former is known as *Consistency Distillation*. To enforce the self-consistency property, we maintain a target model $\boldsymbol{\theta}^-$, updated with exponential moving average (EMA) of the parameter $\boldsymbol{\theta}$ we intend to learn, i.e., $\boldsymbol{\theta}^- \leftarrow \mu\boldsymbol{\theta}^- + (1 - \mu)\boldsymbol{\theta}$, and define the consistency loss as follows:

$$\mathcal{L}(\boldsymbol{\theta}, \boldsymbol{\theta}^-; \Phi) = \mathbb{E}_{\boldsymbol{x}, t} \left[ d \left( \boldsymbol{f}_{\boldsymbol{\theta}}(\boldsymbol{x}_{t_{n+1}}, t_{n+1}), \boldsymbol{f}_{\boldsymbol{\theta}^-}(\hat{\boldsymbol{x}}_{t_n}^{\phi}, t_n) \right) \right], \tag{6}$$

where $d(\cdot, \cdot)$ is a chosen metric function for measuring the distance between two samples, e.g., the squared $\ell_2$ distance $d(\boldsymbol{x}, \boldsymbol{y}) = \|\boldsymbol{x} - \boldsymbol{y}\|_2^2$. $\hat{\boldsymbol{x}}_{t_n}^{\phi}$ is a one-step estimation of $\boldsymbol{x}_{t_n}$ from $\boldsymbol{x}_{t_{n+1}}$ as:

$$\hat{\boldsymbol{x}}_{t_n}^{\phi} \leftarrow \boldsymbol{x}_{t_{n+1}} + (t_n - t_{n+1})\Phi(\boldsymbol{x}_{t_{n+1}}, t_{n+1}; \phi). \tag{7}$$

where $\Phi$ denotes the one-step ODE solver applied to PF-ODE in Eq. 24. (Song et al., 2023) used Euler (Song et al., 2020b) or Heun solver (Karras et al., 2022) as the numerical ODE solver. More details and the pseudo-code for consistency distillation (Algorithm 2) are provided in Appendix A.

# 4 LATENT CONSISTENCY MODELS

Consistency Models (CMs) (Song et al., 2023) only focused on image generation tasks on ImageNet 64×64 (Deng et al., 2009) and LSUN 256×256 (Yu et al., 2015). The potential of CMs to generate higher-resolution text-to-image tasks remains unexplored. In this paper, we introduce **Latent Consistency Models** (LCMs) in Sec 4.1 to tackle these more challenging tasks, unleashing the potential of CMs. Similar to LDMs, our LCMs adopt a consistency model in the image latent space. We choose the powerful Stable Diffusion (SD) as the underlying diffusion model to distill from. We aim to achieve few-step (2∼4) and even one-step inference on SD without compromising image quality. The classifier-free guidance (CFG) (Ho & Salimans, 2022) is an effective technique to further improve sample quality and is widely used in SD. However, its application in CMs remains unexplored. We propose a simple one-stage guided distillation method in Sec 4.2 that solves an *augmented PF-ODE*, integrating CFG into LCM effectively. We propose SKIPPING-STEP technique to accelerate the convergence of LCMs in Sec. 4.3. Finally, we propose Latent Consistency Fine-tuning to finetune a pre-trained LCM for few-step inference on a customized dataset in Sec 4.4.

## 4.1 CONSISTENCY DISTILLATION IN THE LATENT SPACE

Utilizing image latent space in large-scale diffusion models like Stable Diffusion (SD) (Rombach et al., 2022) has effectively enhanced image generation quality and reduced computational load. In SD, an autoencoder $(\mathcal{E}, \mathcal{D})$ is first trained to compress high-dim image data into low-dim latent vector $z = \mathcal{E}(x)$, which is then decoded to reconstruct the image as $\hat{x} = \mathcal{D}(z)$. Training diffusion models in the latent space greatly reduces the computation costs compared to pixel-based models and speeds up the inference process; LDMs make it possible to generate high-resolution images on laptop GPUs. For LCMs, we leverage the advantage of the latent space for consistency distillation, contrasting with the pixel space used in CMs (Song et al., 2023). This approach, termed **Latent Consistency Distillation (LCD)** is applied to pre-trained SD, allowing the synthesis of high-resolution (e.g., 768×768) images in 1∼4 steps. We focus on conditional generation. Recall that the PF-ODE of the reverse diffusion process (Song et al., 2020b; Lu et al., 2022a) is

$$\frac{\mathrm{d}\boldsymbol{z}_t}{\mathrm{d}t} = f(t)\boldsymbol{z}_t + \frac{g^2(t)}{2\sigma_t}\epsilon_{\boldsymbol{\theta}}\left(\boldsymbol{z}_t, \boldsymbol{c}, t\right), \quad \boldsymbol{z}_T \sim \mathcal{N}\left(\boldsymbol{0}, \tilde{\sigma}^2\boldsymbol{I}\right), \tag{8}$$

where $z_t$ are image latents, $\epsilon_\theta\left(z_t, c, t\right)$ is the noise prediction model, and $c$ is the given condition (e.g text). Samples can be drawn by solving the PF-ODE from $T$ to 0. To perform **LCD**, we introduce the consistency function $f_\theta : (z_t, c, t) \mapsto z_0$ to directly predict the solution of *PF-ODE* (Eq. 8) for $t = 0$. We parameterize $f_\theta$ by the noise prediction model $\hat{\epsilon}_\theta$, as follows:

$$f_\theta(z, c, t) = c_{\text{skip}}(t)z + c_{\text{out}}(t)\left(\frac{z - \sigma_t\hat{\epsilon}_\theta(z, c, t)}{\alpha_t}\right), \quad (\epsilon\text{-Prediction}) \tag{9}$$

where $c_{\text{skip}}(0) = 1, c_{\text{out}}(0) = 0$ and $\hat{\epsilon}_\theta(z, c, t)$ is a noise prediction model that initializes with the same parameters as the teacher diffusion model. Notably, $f_\theta$ can be parameterized in various ways, depending on the teacher diffusion model parameterizations of predictions (e.g., $x$, $\epsilon$ (Ho et al., 2020), $v$ (Salimans & Ho, 2022)). We discuss other possible parameterizations in Appendix D.

We assume that an efficient ODE solver $\Psi(z_t, t, s, c)$ is available for approximating the integration of the right-hand side of Eq equation 8 from time $t$ to $s$. In practice, we can use DDIM (Song et al., 2020a), DPM-Solver (Lu et al., 2022a) or DPM-Solver++ (Lu et al., 2022b) as $\Psi(\cdot, \cdot, \cdot, \cdot)$. Note that we only use these solvers in training/distillation, not in inference. We will discuss these solvers further when we introduce the SKIPPING-STEP technique in Sec. 4.3. LCM aims to predict the solution of the PF-ODE by minimizing the consistency distillation loss (Song et al., 2023):

$$\mathcal{L}_{\mathcal{CD}}\left(\theta, \theta^-; \Psi\right) = \mathbb{E}_{z, c, n}\left[d\left(f_\theta(z_{t_{n+1}}, c, t_{n+1}), f_{\theta^-}(\hat{z}_{t_n}^\Psi, c, t_n)\right)\right]. \tag{10}$$

Here, $\hat{z}_{t_n}^\Psi$ is an estimation of the evolution of the *PF-ODE* from $t_{n+1} \rightarrow t_n$ using ODE solver $\Psi$:

$$\hat{z}_{t_n}^\Psi - z_{t_{n+1}} = \int_{t_{n+1}}^{t_n}\left(f(t)z_t + \frac{g^2(t)}{2\sigma_t}\epsilon_\theta\left(z_t, c, t\right)\right)\mathrm{d}t \approx \Psi(z_{t_{n+1}}, t_{n+1}, t_n, c), \tag{11}$$

where the solver $\Psi(\cdot, \cdot, \cdot, \cdot)$ is used to approximate the integration from $t_{n+1} \rightarrow t_n$.

## 4.2 ONE-STAGE GUIDED DISTILLATION BY SOLVING AUGMENTED PF-ODE

Classifier-free guidance (CFG) (Ho & Salimans, 2022) is crucial for synthesizing high-quality text-aligned images in SD, typically needing a CFG scale $\omega$ over 6. Thus, integrating CFG into a distillation method becomes indispensable. Previous method Guided-Distill (Meng et al., 2023) introduces a two-stage distillation to support few-step sampling from a guided diffusion model. However, it is computationally intensive (e.g. at least **45** A100 GPUs **Days** for 2-step inference, estimated in (Liu et al., 2023)). An LCM demands merely **32** A100 GPUs **Hours** training for 2-step inference, as depicted in Figure 1. Furthermore, the two-stage guided distillation might result in accumulated error, leading to suboptimal performance. In contrast, LCMs adopt efficient one-stage guided distillation by solving an augmented PF-ODE. Recall the CFG used in reverse diffusion process:

$$\tilde{\epsilon}_\theta\left(z_t, \omega, c, t\right) := (1 + \omega)\epsilon_\theta\left(z_t, c, t\right) - \omega\epsilon_\theta\left(z_t, \varnothing, t\right), \tag{12}$$

where the original noise prediction is replaced by the linear combination of conditional and unconditional noise and $\omega$ is called the *guidance scale*. To sample from the guided reverse process, we need to solve the following *augmented PF-ODE*: (i.e., augmented with the terms related to $\omega$)

$$\frac{\mathrm{d}z_t}{\mathrm{d}t} = f(t)z_t + \frac{g^2(t)}{2\sigma_t}\tilde{\epsilon}_\theta\left(z_t, \omega, c, t\right), \quad z_T \sim \mathcal{N}\left(0, \tilde{\sigma}^2 I\right). \tag{13}$$

To efficiently perform one-stage guided distillation, we introduce an *augmented consistency function* $f_\theta : (z_t, \omega, c, t) \mapsto z_0$ to directly predict the solution of *augmented PF-ODE* (Eq. 13) for $t = 0$. We parameterize the $f_\theta$ in the same way as in Eq. 9, except that $\hat{\epsilon}_\theta(z, c, t)$ is replaced by $\hat{\epsilon}_\theta(z, \omega, c, t)$, which is a noise prediction model initializing with the same parameters as the teacher diffusion model, but also contains additional trainable parameters for conditioning on $\omega$. The consistency loss is the same as Eq. 10 except that we use augmented consistency function $f_\theta(z_t, \omega, c, t)$.

$$\mathcal{L}_{\mathcal{CD}}\left(\theta, \theta^-; \Psi\right) = \mathbb{E}_{z, c, \omega, n}\left[d\left(f_\theta(z_{t_{n+1}}, \omega, c, t_{n+1}), f_{\theta^-}(\hat{z}_{t_n}^{\Psi, \omega}, \omega, c, t_n)\right)\right] \tag{14}$$

In Eq 14, $\omega$ and $n$ are uniformly sampled from interval $[\omega_{\min}, \omega_{\max}]$ and $\{1, \ldots, N-1\}$ respectively. $\hat{z}_{t_n}^{\Psi, \omega}$ is estimated using the new noise model $\tilde{\epsilon}_\theta\left(z_t, \omega, c, t\right)$, as follows:

$$\begin{aligned}
\hat{z}_{t_n}^{\Psi, \omega} - z_{t_{n+1}} &= \int_{t_{n+1}}^{t_n}\left(f(t)z_t + \frac{g^2(t)}{2\sigma_t}\tilde{\epsilon}_\theta\left(z_t, \omega, c, t\right)\right)\mathrm{d}t \\
&= (1 + \omega)\int_{t_{n+1}}^{t_n}\left(f(t)z_t + \frac{g^2(t)}{2\sigma_t}\epsilon_\theta\left(z_t, c, t\right)\right)\mathrm{d}t - \omega\int_{t_{n+1}}^{t_n}\left(f(t)z_t + \frac{g^2(t)}{2\sigma_t}\epsilon_\theta\left(z_t, \varnothing, t\right)\right)\mathrm{d}t \\
&\approx (1 + \omega)\Psi(z_{t_{n+1}}, t_{n+1}, t_n, c) - \omega\Psi(z_{t_{n+1}}, t_{n+1}, t_n, \varnothing).
\end{aligned} \tag{15}$$

Again, we can use DDIM (Song et al., 2020a), DPM-Solver (Lu et al., 2022a) or DPM-Solver++ (Lu et al., 2022b) as the PF-ODE solver $\Psi(\cdot, \cdot, \cdot, \cdot)$.

## 4.3 Accelerating Distillation with Skipping Time Steps

Discrete diffusion models (Ho et al., 2020; Song & Ermon, 2019) typically train noise prediction models with a long time-step schedule $\{t_i\}_i$ (also called discretization schedule or time schedule) to achieve high quality generation results. For instance, Stable Diffusion (SD) has a time schedule of length 1,000. However, directly applying Latent Consistency Distillation (**LCD**) to SD with such an extended schedule can be problematic. The model needs to sample across all 1,000 time steps, and the consistency loss attempts to aligns the prediction of LCM model $f_\theta(z_{t_{n+1}}, c, t_{n+1})$ with the prediction $f_\theta(z_{t_n}, c, t_n)$ at the subsequent step along the same trajectory. Since $t_n - t_{n+1}$ is tiny, $z_{t_n}$ and $z_{t_{n+1}}$ (and thus $f_\theta(z_{t_{n+1}}, c, t_{n+1})$ and $f_\theta(z_{t_n}, c, t_n)$) are already close to each other, incurring small consistency loss and hence leading to slow convergence. To address this issues, we introduce the Skipping-Step method to considerably shorten the length of time schedule (from thousands to dozens) to achieve fast convergence while preserving generation quality.

Consistency Models (CMs) (Song et al., 2023) use the EDM (Karras et al., 2022) continuous time schedule, and the Euler, or Heun Solver as the numerical continuous PF-ODE solver. For LCMs, in order to adapt to the discrete-time schedule in Stable Diffusion, we utilize DDIM (Song et al., 2020a), DPM-Solver (Lu et al., 2022a), or DPM-Solver++ (Lu et al., 2022b) as the ODE solver. (Lu et al., 2022a) shows that these advanced solvers can solve the PF-ODE efficiently in Eq. 8. Now, we introduce the Skipping-Step method in Latent Consistency Distillation (LCD). Instead of ensuring consistency between adjacent time steps $t_{n+1} \to t_n$, LCMs aim to ensure consistency between the current time step and $k$-step away, $t_{n+k} \to t_n$. Note that setting $k$=1 reduces to the original schedule in (Song et al., 2023), leading to slow convergence, and very large $k$ may incur large approximation errors of the ODE solvers. In our main experiments, we set $k$=20, drastically reducing the length of time schedule from thousands to dozens. Results in Sec. 5.2 show the effect of various $k$ values and reveal that the Skipping-Step method is crucial in accelerating the LCD process. Specifically, consistency distillation loss in Eq. 14 is modified to ensure consistency from $t_{n+k}$ to $t_n$:

$$\mathcal{L}_{\mathcal{CD}}\left(\theta, \theta^-; \Psi\right) = \mathbb{E}_{z,c,\omega,n}\left[d\left(f_\theta(z_{t_{n+k}}, \omega, c, t_{n+k}), f_{\theta^-}(\hat{z}_{t_n}^{\Psi,\omega}, \omega, c, t_n)\right)\right], \tag{16}$$

with $\hat{z}_{t_n}^{\Psi,\omega}$ being an estimate of $z_{t_n}$ using numerical *augmented PF-ODE* solver $\Psi$:

$$\hat{z}_{t_n}^{\Psi,\omega} \longleftarrow z_{t_{n+k}} + (1+\omega)\Psi(z_{t_{n+k}}, t_{n+k}, t_n, c) - \omega\Psi(z_{t_{n+k}}, t_{n+k}, t_n, \varnothing). \tag{17}$$

The above derivation is similar to Eq. 15. For LCM, we use three possible ODE solvers here: DDIM (Song et al., 2020a), DPM-Solver (Lu et al., 2022a), DPM-Solver++ (Lu et al., 2022b), and we compare their performance in Sec 5.2. In fact, DDIM (Song et al., 2020a) is the first-order discretization approximation of the DPM-Solver (Proven in (Lu et al., 2022a)). Here we provide the detailed formula of the DDIM PF-ODE solver $\Psi_{\text{DDIM}}$ from $t_{n+k}$ to $t_n$. The formulas of the other two solver $\Psi_{\text{DPM-Solver}}$, $\Psi_{\text{DPM-Solver++}}$ are provided in Appendix E.

$$\Psi_{\text{DDIM}}(z_{t_{n+k}}, t_{n+k}, t_n, c) = \underbrace{\frac{\alpha_{t_n}}{\alpha_{t_{n+k}}}z_{t_{n+k}} - \sigma_{t_n}\left(\frac{\sigma_{t_{n+k}} \cdot \alpha_{t_n}}{\alpha_{t_{n+k}} \cdot \sigma_{t_n}} - 1\right)\hat{\epsilon}_\theta(z_{t_{n+k}}, c, t_{n+k})}_{\text{DDIM Estimated } z_{t_n}} - z_{t_{n+k}} \tag{18}$$

We present the pseudo-code for **LCD** with CFG and Skipping-Step techniques in Algorithm 1 The modifications from the original Consistency Distillation (CD) algorithm in Song et al. (2023) are highlighted in blue. We use $\ell_2$ norm for distance metric. Also, the LCM sampling algorithm 3 is provided in Appendix B.

---

**Algorithm 1** Latent Consistency Distillation (LCD)

---

**Input:** dataset $\mathcal{D}$, initial model parameter $\theta$, learning rate $\eta$, ODE solver $\Psi(\cdot,\cdot,\cdot,\cdot)$, distance metric $d(\cdot,\cdot)$, EMA rate $\mu$, noise schedule $\alpha(t), \sigma(t)$, guidance scale $[w_{\min}, w_{\max}]$, skipping interval $k$, and encoder $E(\cdot)$
Encoding training data into latent space: $\mathcal{D}_z = \{(z, c)|z = E(x), (x, c) \in \mathcal{D}\}$
$\theta^- \leftarrow \theta$
**repeat**
    Sample $(z, c) \sim \mathcal{D}_z$, $n \sim \mathcal{U}[1, N-k]$ and $\omega \sim [\omega_{\min}, \omega_{\max}]$
    Sample $z_{t_{n+k}} \sim \mathcal{N}(\alpha(t_{n+k})z; \sigma^2(t_{n+k})\mathbf{I})$
    $\hat{z}_{t_n}^{\Psi,\omega} \leftarrow z_{t_{n+k}} + (1+\omega)\Psi(z_{t_{n+k}}, t_{n+k}, t_n, c) - \omega\Psi(z_{t_{n+k}}, t_{n+k}, t_n, \varnothing)$
    $\mathcal{L}(\theta, \theta^-; \Psi) \leftarrow d(f_\theta(z_{t_{n+k}}, \omega, c, t_{n+k}), f_{\theta^-}(\hat{z}_{t_n}^{\Psi,\omega}, \omega, c, t_n))$
    $\theta \leftarrow \theta - \eta\nabla_\theta\mathcal{L}(\theta, \theta^-)$
    $\theta^- \leftarrow \text{stopgrad}(\mu\theta^- + (1-\mu)\theta)$
**until** convergence

---

| Model ($512 \times 512$) Reso | FID $\downarrow$ | | | | CLIP Score $\uparrow$ | | | |
|---|---|---|---|---|---|---|---|---|
| | 1 Step | 2 Steps | 4 Steps | 8 Steps | 1 Steps | 2 Steps | 4 Steps | 8 Steps |
| DDIM (Song et al., 2020a) | 183.29 | 81.05 | 22.38 | 13.83 | 6.03 | 14.13 | 25.89 | 29.29 |
| DPM (Lu et al., 2022a) | 185.78 | 72.81 | 18.53 | 12.24 | 6.35 | 15.10 | 26.64 | 29.54 |
| DPM++ (Lu et al., 2022b) | 185.78 | 72.81 | 18.43 | 12.20 | 6.35 | 15.10 | 26.64 | **29.55** |
| Guided-Distill (Meng et al., 2023) | 108.21 | 33.25 | 15.12 | 13.89 | 12.08 | 22.71 | 27.25 | 28.17 |
| LCM (Ours) | **35.36** | **13.31** | **11.10** | 11.84 | **24.14** | **27.83** | **28.69** | 28.84 |

Table 1: Quantitative results with $\omega = 8$ at 512×512 resolution. LCM significantly surpasses baselines in the 1-4 step region on LAION-Aesthetic-6+ dataset. For LCM, DDIM-Solver is used with a skipping step of $k = 20$. For reference, when using DDIM with 50 steps, the FID is 10.74, and the CLIP score is 30.34.

| Model ($768 \times 768$) Reso | FID $\downarrow$ | | | | CLIP Score $\uparrow$ | | | |
|---|---|---|---|---|---|---|---|---|
| | 1 Step | 2 Steps | 4 Steps | 8 Steps | 1 Steps | 2 Steps | 4 Steps | 8 Steps |
| DDIM (Song et al., 2020a) | 186.83 | 77.26 | 24.28 | 15.66 | 6.93 | 16.32 | 26.48 | 29.49 |
| DPM (Lu et al., 2022a) | 188.92 | 67.14 | 20.11 | **14.08** | 7.40 | 17.11 | 27.25 | 29.80 |
| DPM++ (Lu et al., 2022b) | 188.91 | 67.14 | 20.08 | 14.11 | 7.41 | 17.11 | 27.26 | **29.84** |
| Guided-Distill (Meng et al., 2023) | 120.28 | 30.70 | 16.70 | 14.12 | 12.88 | 24.88 | 28.45 | 29.16 |
| LCM (Ours) | **34.22** | **16.32** | **13.53** | 14.97 | **25.32** | **27.92** | **28.60** | 28.49 |

Table 2: Quantitative results with $\omega = 8$ at 768×768 resolution. LCM significantly surpasses the baselines in the 1-4 step region on LAION-Aesthetic-6.5+ dataset. For LCM, DDIM-Solver is used with a skipping step of $k = 20$. For reference, when using DDIM with 50 steps, the FID is 12.74, and the CLIP score is 30.82.

### 4.4 Latent Consistency Fine-tuning for customized dataset

Foundation generative models like Stable Diffusion excel in diverse text-to-image generation tasks but often require fine-tuning on customized datasets to meet the requirements of downstream tasks. We propose Latent Consistency Fine-tuning (LCF), a fine-tuning method for pretrained LCM. Inspired by Consistency Training (CT) (Song et al., 2023), LCF enables efficient few-step inference on customized datasets without relying on a teacher diffusion model trained on such data. This approach presents a viable alternative to traditional fine-tuning methods for diffusion models. The pseudo-code for LCF is provided in Algorithm 4, with a more detailed illustration in Appendix C.

## 5 Experiment

In this section, we employ latency consistency distillation to train LCM on two subsets of LAION-5B. In Sec 5.1, we first evaluate the performance of LCM on text-to-image generation tasks. In Sec 5.2, we provide a detailed ablation study to test the effectiveness of using different solvers, skipping step schedules and guidance scales. Lastly, in Sec 5.3, we present the experimental results of latent consistency finetuning on a pretrained LCM on customized image datasets.

### 5.1 Text-to-Image Generation

**Datasets** We use two subsets of LAION-5B (Schuhmann et al., 2022): LAION-Aesthetics-6+ (12M) and LAION-Aesthetics-6.5+ (650K) for text-to-image generation. Our experiments consider resolutions of 512×512 and 768×768. For 512 resolution, we use LAION-Aesthetics-6+, which comprises 12M text-image pairs with predicted aesthetics scores higher than 6. For 768 resolution, we use LAION-Aesthetics-6.5+, with 650K text-image pairs with aesthetics score higher than 6.5.

**Model Configuration** For 512 resolution, we use the pre-trained Stable Diffusion-V2.1-Base (Rombach et al., 2022) as the teacher model, which was originally trained on resolution 512×512 with $\epsilon$-Prediction (Ho et al., 2020). For 768 resolution, we use the widely used pre-trained Stable Diffusion-V2.1, originally trained on resolution 768×768 with $v$-Prediction (Salimans & Ho, 2022). We train LCM with 100K iterations and we use a batch size of 72 for $(512 \times 512)$ setting, and 16 for $(768 \times 768)$ setting, the same learning rate 8e-6 and EMA rate $\mu = 0.999943$ as used in (Song et al., 2023). For *augmented PF-ODE* solver $\Psi$ and skipping step $k$ in Eq. 17, we use DDIM-Solver (Song et al., 2020a) with skipping step $k = 20$. We set the guidance scale range $[w_{\min}, w_{\max}] = [2, 14]$, consistent with (Meng et al., 2023). More training details are provided in the Appendix F.

**Baselines & Evaluation** We use DDIM (Song et al., 2020a), DPM (Lu et al., 2022a), DPM++ (Lu et al., 2022b) and Guided-Distill (Meng et al., 2023) as baselines. The first three are training-free samplers requiring more peak memory per step with classifier-free guidance. Guided-Distill requires two stages of guided distillation. Since Guided-Distill is not open-sourced, we strictly followed the training procedure outlined in the paper to reproduce the results. Due to the limited resource (Meng et al. (2023) used a large batch size of 512, requiring at least 32 A100 GPUs), we reduce the batch size to 72, the same as ours, and trained for the same 100K iterations. Reproduction details are

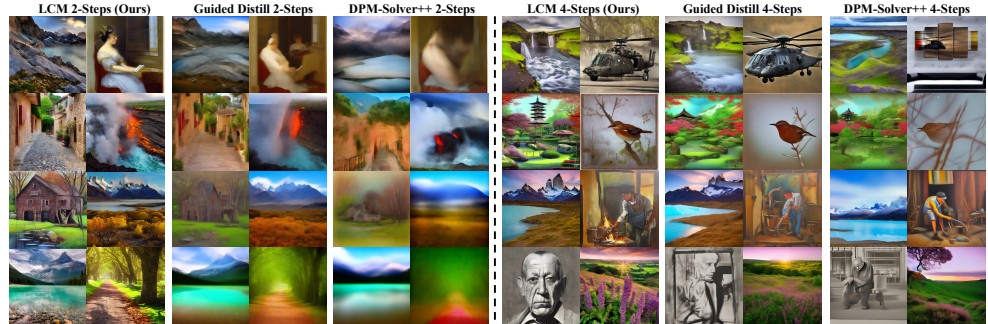

Figure 2: Text-to-Image generation results on LAION-Aesthetic-6.5+ with 2-, 4-step inference. Images generated by LCM exhibit superior detail and quality, outperforming other baselines by a large margin.

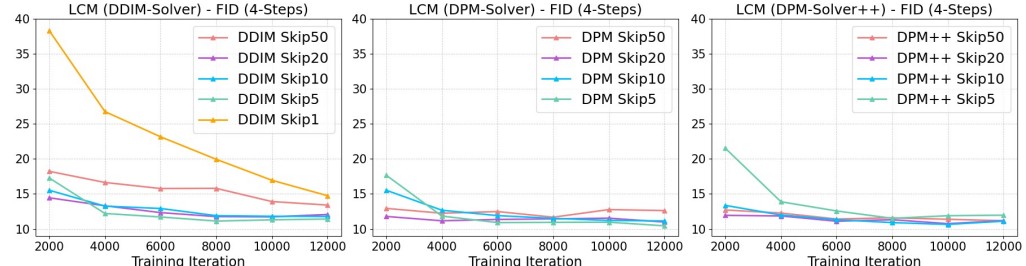

Figure 3: Ablation study on different ODE solvers and skipping step $k$. Appropriate skipping step $k$ can significantly accelerate convergence and lead to better FID within the same number of training steps.

provided in Appendix G. We admit that longer training and more computational resources can lead to better results as reported in (Meng et al., 2023). However, LCM achieves faster convergence and superior results under the same computation cost. For evaluation, We generate 30K images from 10K text prompts in the test set (3 images per prompt), and adopt FID and CLIP scores to evaluate the diversity and quality of the generated images. We use ViT-g/14 for evaluating CLIP scores.

**Results.** The quantitative results in Tables 1 and 2 show that LCM notably outperforms baseline methods at $512$ and $768$ resolutions, especially in the low step regime (1~4), highlighting its efficiency and superior performance. Unlike DDIM, DPM, DPM++, which require more peak memory per sampling step with CFG, LCM requires only one forward pass per sampling step, saving both time and memory. Moreover, in contrast to the two-stage distillation procedure employed in Guided-Distill, LCM only needs one-stage guided distillation, which is much simpler and more practical. The **qualitative results** in Figure 2 further show the superiority of LCM with 2- and 4-step inference.

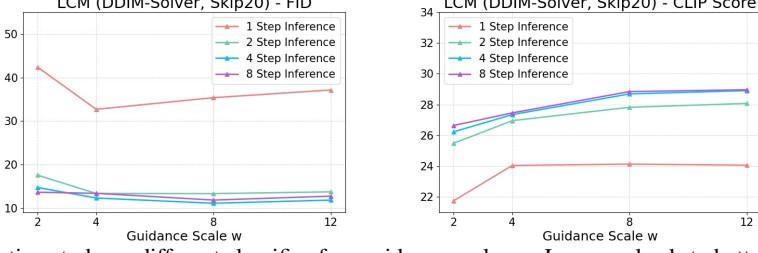

Figure 4: Ablation study on different classifier-free guidance scales $\omega$. Larger $\omega$ leads to better sample quality (CLIP Scores). The performance gaps across 2, 4, and 8 steps are minimal, showing the efficacy of LCM.

## 5.2 ABLATION STUDY

**ODE Solvers & Skipping-Step Schedule.** We compare various solvers $\Psi$ (DDIM (Song et al., 2020a), DPM (Lu et al., 2022a), DPM++ (Lu et al., 2022b)) for solving the *augmented PF-ODE* specified in Eq 17, and explore different skipping step schedules with different $k$. The results are depicted in Figure 3. We observe that: 1) Using SKIPPING-STEP techniques (see Sec 4.3), LCM achieves fast convergence within 2,000 iterations in the 4-step inference setting. Specifically, the DDIM solver converges slowly at skipping step $k = 1$, while setting $k = 5, 10, 20$ leads to much faster convergence, underscoring the effectiveness of the Skipping-Step method. 2) DPM and DPM++ solvers perform better at a larger skipping step ($k = 50$) compared to the DDIM solver which suffers from increased ODE approximation error with larger $k$. This phenomenon is also discussed in (Lu et al., 2022a). 3) Very small $k$ values (1 or 5) result in slow convergence and very large

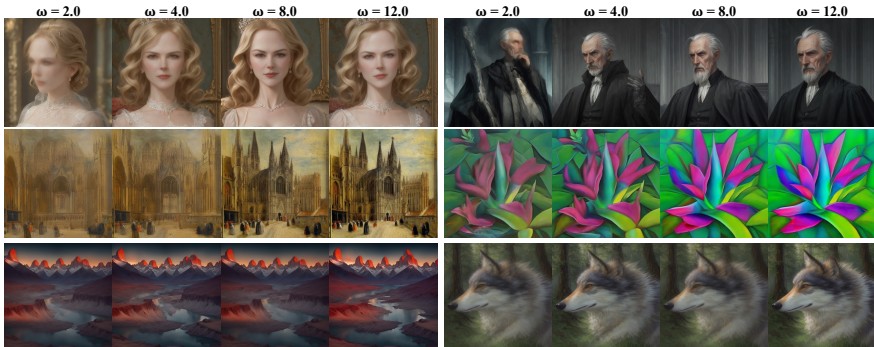

Figure 5: 4-step LCMs using different CFG scales $\omega$. LCMs utilize one-stage guided distillation to directly incorporate CFG scales $\omega$. Larger $\omega$ enhances image quality.

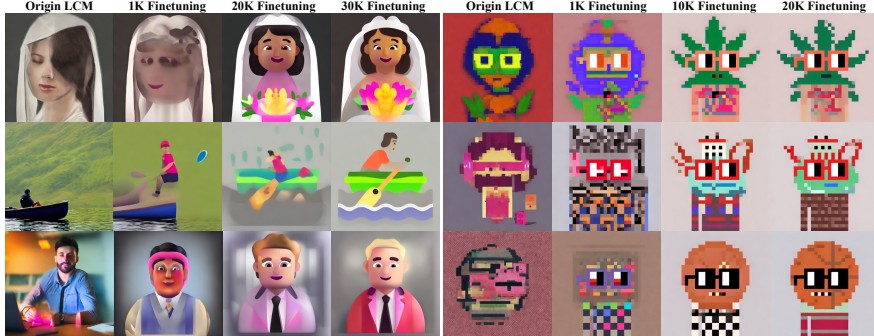

Figure 6: 4-step LCMs using Latent Consistency Fine-tuning (LCF) on two customized datasets: Emoji dataset (left) and Pixel art dataset (right). Through LCF, LCM produces images with customized styles.

ones (e.g., 50 for DDIM) may lead to inferior results. Hence, we choose $k = 20$, which provides competitive performance for all three solvers, for our main experiment in Sec 5.1.

**The Effect of Guidance Scale $\omega$.** We examine the effect of using different CFG scales $\omega$ in LCM. Typically, $\omega$ balances sample quality and diversity. A larger $\omega$ generally tends to improve sample quality (indicated by CLIP), but may compromise diversity (measured by FID). Beyond a certain threshold, an increased $\omega$ yields better CLIP scores at the expense of FID. Figure 4 presents the results for various $\omega$ across different inference steps. Our findings include: 1) Using large $\omega$ enhances sample quality (CLIP Scores) but results in relatively inferior FID. 2) The performance gaps across 2, 4, and 8 inference steps are negligible, highlighting LCM's efficacy in 2~8 step regions. However, a noticeable gap exists in one-step inference, indicating rooms for further improvements. We present visualizations for different $\omega$ in Figure 5. One can see clearly that a larger $\omega$ enhances sample quality, verifying the effectiveness of our one-stage guided distillation method.

### 5.3 DOWNSTREAM CONSISTENCY FINE-TUNING RESULTS

We perform **Latent Consistency Fine-tuning** (LCF) on several customized image datasets, Emoji dataset (Adler, 2022), Pixel art dataset (Piedrafita, 2022), and Pokemon dataset (Pinkney, 2022), to demonstrate the efficiency of LCF. Each dataset, comprised of hundreds of customized text-image pairs, is split such that 90% is used for fine-tuning and the rest 10% for testing. For LCF, we utilize pretrained LCM that was originally trained at the resolution of $768 \times 768$ used in Table 2. For these two datasets, we fine-tune the pre-trained LCM for 30K iterations with a learning rate 8e-6. We present qualitative results of adopting LCF on two customized image datasets in Figure 6. The finetuned LCM is capable of generating images with customized styles in few steps, showing the effectiveness of our method.

### 6 CONCLUSION

We present Latent Consistency Models (LCMs), and a highly efficient one-stage guided distillation method that enables few-step or even one-step inference on pre-trained LDMs. Furthermore, we present latent consistency fine-tuning (LCF), to enable few-step inference of LCMs on customized image datasets. Extensive experiments on the LAION-5B-Aesthetics dataset demonstrate the superior performance and efficiency of LCMs. Future work include extending our method to more image generation tasks such as text-guided image editing, inpainting and super-resolution.

## REPRODUCIBILITY STATEMENT

In our paper, we discuss the data, model, training hyper-parameters as detailed in Section 5.1, Appendix F. Since our approach is straightforward and computation efficient, it ensures a high level of reproducibility of our work.

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

# A    MORE DETAILS ON DIFFUSION AND CONSISTENCY MODELS

## A.1    DIFFUSION MODELS

Consider the forward process, described by the following SDE for $t \in [0, T]$:

$$\mathrm{d}\boldsymbol{x}_t = f(t)\boldsymbol{x}_t \, \mathrm{d}t + g(t)\mathrm{d}\boldsymbol{w}_t, \quad \boldsymbol{x}_0 \sim p_{data}(\boldsymbol{x}_0), \tag{19}$$

where $\boldsymbol{w}_t$ denotes the standard Brownian motion. Leveraging the classic result of Anderson (1982), Song et al. (2020b) show that the reverse process of the above forward process is also a diffusion process, specified by the following reverse-time SDE:

$$\mathrm{d}\boldsymbol{x}_t = \left[ f(t)\boldsymbol{x}_t - g^2(t)\nabla_{\boldsymbol{x}} \log q_t(\boldsymbol{x}_t) \right] \mathrm{d}t + g(t)\mathrm{d}\overline{\boldsymbol{w}}_t, \quad \boldsymbol{x}_T \sim q_T(\boldsymbol{x}_T), \tag{20}$$

where $\overline{\boldsymbol{w}}_t$ is a standard reverse-time Brownian motion. One can leverage the reverse SDE for data sampling from $T$ to 0, starting with $q_T(\boldsymbol{x}_T)$, which follows a Gaussian distribution approximately. However, directly sampling from the reverse SDE requires a large number of discretization steps and is typically very slow. To accelerate the sampling process, prior work (e.g., (Song et al., 2020b; Lu et al., 2022a) leveraged the relation between the above SDE and ODE and designed ODE solvers for sampling. In particular, it is known that for SDE (Eq.20), the following ordinary differential equation (ODE), called the *Probability Flow ODE* (PF-ODE), has the same marginal distribution $q_t(\boldsymbol{x})$ (Song et al., 2020b; Lu et al., 2022a):

$$\frac{\mathrm{d}\boldsymbol{x}_t}{\mathrm{d}t} = f(t)\boldsymbol{x}_t - \frac{1}{2}g^2(t)\nabla_{\boldsymbol{x}} \log q_t(\boldsymbol{x}_t), \quad \boldsymbol{x}_T \sim q_T(\boldsymbol{x}_T) \tag{21}$$

The term $-\nabla \log q_t(\boldsymbol{x}_t)$ in Eq. 21 is typically called the *score function* of $q_t(\boldsymbol{x}_t)$. In diffusion models, we train the noise prediction model $\boldsymbol{\epsilon}_\theta(\boldsymbol{x}_t, t)$ to fit the scaled score function, via minimizing the following score matching objective:

$$\mathcal{L}(\theta) = \mathbb{E}_{t \in [0,T], x_t \sim q_t} \left[ w(t)||\boldsymbol{\epsilon}_\theta(\boldsymbol{x}_t, t) + \sigma(t)\nabla \log q_t(\boldsymbol{x}_t))||^2 \right]$$
$$= \mathbb{E}_{t \in [0,T], \boldsymbol{x}_0 \sim q_0, \boldsymbol{\epsilon}} \left[ w(t)||\boldsymbol{\epsilon}_\theta(\boldsymbol{x}_t, t) - \boldsymbol{\epsilon}||^2 \right] \tag{22}$$

where $w(t)$ is the weight function, $\boldsymbol{\epsilon} \sim N(0, I)$ and $\boldsymbol{x}_t = \alpha(t)\boldsymbol{x}_0 + \sigma(t)\boldsymbol{\epsilon}$. By substituting the score function with the noise prediction model in Eq. 21, we obtain the following ODE, which can be used for sampling:

$$\frac{\mathrm{d}\boldsymbol{x}_t}{\mathrm{d}t} = f(t)\boldsymbol{x}_t + \frac{g^2(t)}{2\sigma_t}\boldsymbol{\epsilon}_\theta(\boldsymbol{x}_t, t), \quad \boldsymbol{x}_T \sim \mathcal{N}(\mathbf{0}, \tilde{\sigma}^2 \boldsymbol{I}). \tag{23}$$

## A.2    MORE DETAILS ON CONSISTENCY MODELS IN (SONG ET AL., 2023)

In this subsection, we provide more details on the consistency models and consistency distillation algorithm in (Song et al., 2023). The pre-trained diffusion model used in (Song et al., 2023) adopts the continuous noise schedule from EDM (Karras et al., 2022), therefore the PF-ODE in Eq. 23 can be simplified as:

$$\frac{\mathrm{d}\mathbf{x}_t}{\mathrm{d}t} = -t\nabla \log q_t(\boldsymbol{x}_t) \approx -t\boldsymbol{s}_\phi(\mathbf{x}_t, t), \tag{24}$$

where the $\boldsymbol{s}_\phi(\mathbf{x}_t, t) \approx \nabla \log q_t(\boldsymbol{x}_t)$ is a score prediction model trained via score matching (Hyvärinen & Dayan, 2005; Song & Ermon, 2019). Note that different noise schedules result in different PF-ODE and the PF-ODE in Eq. 24 corresponds to the EDM noise schedule (Karras et al., 2022). We denote the one-step ODE solver applied to PF-ODE in Eq. 24 as $\Phi(\boldsymbol{x}_t, t; \phi)$. One can either use Euler (Song et al., 2020b) or Heun solver (Karras et al., 2022) as the numerical ODE solver. Then, we use the ODE solver to estimate the evolution of a sample $\boldsymbol{x}_{t_n}$ from $\boldsymbol{x}_{t_{n+1}}$ as:

$$\hat{\boldsymbol{x}}_{t_n}^\phi \leftarrow \boldsymbol{x}_{t_{n+1}} + (t_n - t_{n+1})\Phi(\boldsymbol{x}_{t_{n+1}}, t_{n+1}; \phi). \tag{25}$$

(Song et al., 2020b) used the same time schedule as in (Karras et al., 2022): $t_i = (\epsilon^{1/\rho} + \frac{i-1}{N-1}(T^{1/\rho} - \epsilon^{1/\rho}))^\rho$, and $\rho = 7$. To enforce the self-consistency property in Eq. 4, we maintain a target model $\boldsymbol{\theta}^-$, which is updated with exponential moving average (EMA) of the parameter $\boldsymbol{\theta}$ we intend to learn, i.e., $\boldsymbol{\theta}^- \leftarrow \mu\boldsymbol{\theta}^- + (1 - \mu)\boldsymbol{\theta}$, and define the consistency loss as follows:

$$\mathcal{L}(\boldsymbol{\theta}, \boldsymbol{\theta}^-; \Phi) = \mathbb{E}_{\boldsymbol{x},t} \left[ d\left( \boldsymbol{f}_{\boldsymbol{\theta}}(\boldsymbol{x}_{t_{n+1}}, t_{n+1}), \boldsymbol{f}_{\boldsymbol{\theta}^-}(\hat{\boldsymbol{x}}_{t_n}^\phi, t_n) \right) \right], \tag{26}$$

where $d(\cdot, \cdot)$ is a chosen metric function for measuring the distance between two samples, e.g., the squared $\ell_2$ distance $d(\boldsymbol{x}, \boldsymbol{y}) = ||\boldsymbol{x} - \boldsymbol{y}||_2^2$. The pseudo-code for consistency distillation in Song et al. (2023). is presented in Algorithm 2. In their original paper, an Euler solver was used as the ODE solver for the continuous-time setting.

---

**Algorithm 2** Consistency Distillation (CD) (Song et al., 2023)

**Input:** dataset $\mathcal{D}$, initial model parameter $\boldsymbol{\theta}$, learning rate $\eta$, ODE solver $\Phi(\cdot, \cdot, \cdot)$, distance metric $d(\cdot, \cdot)$, and EMA rate $\mu$
$\boldsymbol{\theta}^- \leftarrow \boldsymbol{\theta}$
**repeat**
    Sample $\boldsymbol{x} \sim \mathcal{D}$ and $n \sim \mathcal{U}[1, N-1]$
    Sample $\boldsymbol{x}_{t_{n+1}} \sim \mathcal{N}(\boldsymbol{x}; t_{n+1}^2\mathbf{I})$
    $\hat{\boldsymbol{x}}_{t_n}^\phi \leftarrow \boldsymbol{x}_{t_{n+1}} + (t_n - t_{n+1})\Phi(\boldsymbol{x}_{t_{n+1}}, t_{n+1}, \phi)$
    $\mathcal{L}(\boldsymbol{\theta}, \boldsymbol{\theta}^-; \Phi) \leftarrow d(\boldsymbol{f}_{\boldsymbol{\theta}}(\boldsymbol{x}_{t_{n+1}}, t_{n+1}), \boldsymbol{f}_{\boldsymbol{\theta}^-}(\hat{\boldsymbol{x}}_{t_n}^\phi, t_n))$
    $\boldsymbol{\theta} \leftarrow \boldsymbol{\theta} - \eta\nabla_{\boldsymbol{\theta}}\mathcal{L}(\boldsymbol{\theta}, \boldsymbol{\theta}^-; \Phi)$
    $\boldsymbol{\theta}^- \leftarrow \text{stopgrad}(\mu\boldsymbol{\theta}^- + (1 - \mu)\boldsymbol{\theta})$
**until** convergence

---

## B   MULTISTEP LATENT CONSISTENCY SAMPLING

Now, we present the multi-step sampling algorithm for latent consistency model. The sampling algorithm for LCM is very similar to the one in consistency models (Song et al., 2023) except the incorporation of classifier-free guidance in LCM. Unlike multi-step sampling in diffusion models, in which we predict $\boldsymbol{z}_{t-1}$ from $\boldsymbol{z}_t$, the latent consistency models directly predicts the origin $\boldsymbol{z}_0$ of augmented PF-ODE trajectory (the solution of the augmented of PF-ODE), given guidance scale $\omega$. This generates samples in a single step. The sample quality can be improved by alternating the denoising and noise injection steps. In particular, in the $n$-th iteration, we first perform noise-injecting forward process to the previous predicted sample $\boldsymbol{z}$ as $\hat{\boldsymbol{z}}_{\tau_n} \sim \mathcal{N}(\alpha(\tau_n)\boldsymbol{z}; \sigma^2(\tau_n)\mathbf{I})$, where $\tau_n$ is a decreasing sequence of time steps. This corresponds to going back to point $\hat{\boldsymbol{z}}_{\tau_n}$ on the PF-ODE trajectory. Then, we perform the next $\boldsymbol{z}_0$ prediction again using the trained latent consistency function. In our experiments, one can see the second iteration can already refine the generation quality significantly, and high quality images can be generated in just 2-4 steps. We provide the pseudo-code in Algorithm 3.

---

**Algorithm 3** Multistep Latent Consistency Sampling

**Input:** Latent Consistency Model $\boldsymbol{f}_{\boldsymbol{\theta}}(\cdot, \cdot, \cdot, \cdot)$, Sequence of timesteps $\tau_1 > \tau_2 > \cdots > \tau_{N-1}$, Text condition $\boldsymbol{c}$, Classifier-Free Guidance Scale $\omega$, Noise schedule $\alpha(t), \sigma(t)$, Decoder $D(\cdot)$
Sample initial noise $\hat{\boldsymbol{z}}_T \sim \mathcal{N}(\mathbf{0}; \boldsymbol{I})$
$\boldsymbol{z} \leftarrow \boldsymbol{f}_{\boldsymbol{\theta}}(\hat{\boldsymbol{z}}_T, \omega, \boldsymbol{c}, T)$
**for** $n = 1$ to $N - 1$ **do**
    $\hat{\boldsymbol{z}}_{\tau_n} \sim \mathcal{N}(\alpha(\tau_n)\boldsymbol{z}; \sigma^2(\tau_n)\mathbf{I})$
    $\boldsymbol{z} \leftarrow \boldsymbol{f}_{\boldsymbol{\theta}}(\hat{\boldsymbol{z}}_{\tau_n}, \omega, \boldsymbol{c}, \tau_n)$
**end for**
$\boldsymbol{x} \leftarrow D(\boldsymbol{z})$
**Output:** $\boldsymbol{x}$

---

## C  ALGORITHM DETAILS OF LATENT CONSISTENCY FINE-TUNING

In this section, we provide further details of Latent Consistency Fine-tuning (LCF). The pseudo-code of LCF is provided in Algorithm 4. During the Latent Consistency Fine-tuning (LCF) process, we randomly select two time steps $t_n$ and $t_{n+k}$ that are $k$ time steps apart and apply the *same* Gaussian noise $\epsilon$ to obtain the noised data $z_{t_n}, z_{t_{n+k}}$ as follows:

$$z_{t_{n+k}} = \alpha(t_{n+k})z + \sigma(t_{n+k})\epsilon \quad , \quad z_{t_n} = \alpha(t_n)z + \sigma(t_n)\epsilon.$$

Then, we can directly calculate the consistency loss for these two time steps to enforce self-consistency property in Eq.4. Notably, this method can also utilize the skipping-step technique to speedup the convergence. Furthermore, we note that latent consistency fine-tuning is independent of the pre-trained teacher model, facilitating direct fine-tuning of a pre-trained latent consistency model without reliance on the teacher diffusion model.

---

**Algorithm 4** Latent Consistency Fine-tuning (LCF)

---

**Input:** customized dataset $\mathcal{D}^{(s)}$, pre-trained LCM parameter $\theta$, learning rate $\eta$, distance metric $d(\cdot, \cdot)$, EMA rate $\mu$, noise schedule $\alpha(t), \sigma(t)$, guidance scale $[w_{\min}, w_{\max}]$, skipping interval $k$, and encoder $E(\cdot)$

Encode training data into the latent space: $\mathcal{D}_z^{(s)} = \{(z, c) | z = E(x), (x, c) \in \mathcal{D}^{(s)}\}$
$\theta^- \leftarrow \theta$
**repeat**
    Sample $(z, c) \sim \mathcal{D}_z^{(s)}, n \sim \mathcal{U}[1, N - k]$ and $w \sim [w_{\min}, w_{\max}]$
    Sample $\epsilon \sim \mathcal{N}(\mathbf{0}, \mathbf{I})$
    $z_{t_{n+k}} \leftarrow \alpha(t_{n+k})z + \sigma(t_{n+k})\epsilon \quad , \quad z_{t_n} \leftarrow \alpha(t_n)z + \sigma(t_n)\epsilon$
    $\mathcal{L}(\theta, \theta^-) \leftarrow d(f_\theta(z_{t_{n+k}}, t_{n+k}, c, w), f_{\theta^-}(z_{t_n}, t_n, c, w))$
    $\theta \leftarrow \theta - \eta \nabla_\theta \mathcal{L}(\theta, \theta^-)$
    $\theta^- \leftarrow \text{stopgrad}(\mu\theta^- + (1 - \mu)\theta)$
**until** convergence

---

## D  DIFFERENT WAYS TO PARAMETERIZE THE CONSISTENCY FUNCTION

As previously discussed in Eq 9, we can parameterize our consistency model function $f_\theta(z, c, t)$ in different ways, depending on the way the teacher diffusion model is parameterized. For $\epsilon$-Prediction (Song et al., 2020a), we use the following parameterization:

$$f_\theta(z, c, t) = c_{\text{skip}}(t)z + c_{\text{out}}(t)\hat{z}_0 \quad (\epsilon\text{-Prediction}) \tag{27}$$

where

$$\hat{z}_0 = \left( \frac{z_t - \sigma(t)\hat{\epsilon}_\theta(z, c, t)}{\alpha(t)} \right). \tag{28}$$

Recalling that $z_t = \alpha(t)z_0 + \sigma(t)\epsilon$, $\hat{z}_0$ can be seen as a prediction of $z_0$ at time $t$.

Next, we provide the parameterization of ($x$-Prediction) (Ho et al., 2020; Salimans & Ho, 2022) with the following form:

$$f_\theta(z, c, t) = c_{\text{skip}}(t)z + c_{\text{out}}(t)x_\theta(z_t, c, t), \quad (x\text{-Prediction}) \tag{29}$$

where $x_\theta(z_t, c, t)$ corresponds to the teacher diffusion model with $x$-prediction.

Finally, for $v$-prediction (Salimans & Ho, 2022), the consistency function is parameterized as

$$f_\theta(z, c, t) = c_{\text{skip}}(t)z + c_{\text{out}}(t)\left(\alpha_t z_t - \sigma_t v_\theta(z_t, c, t)\right), \quad (v\text{-Prediction}) \tag{30}$$

where $v_\theta(z_t, c, t)$ corresponds to the teacher diffusion model with $v$-prediction.

As mentioned in Sec 5.1, we use the $\epsilon$-Parameterization in Eq. 27 to train LCM at 512×512 resolution using the teacher diffusion model, Stable-Diffusion-V2.1-Base (originally trained with $\epsilon$-Prediction at 512 resolution). For resolution 768×768, we train the LCM using the $v$-Parameterization in Eq. 30, adopting the teacher diffusion model, Stable-Diffusion-V2.1 (originally trained with $v$-Prediction at 768 resolution).

## E    FORMULAS OF OTHER ODE SOLVERS

As discussed in Sec 4.3, we use the DDIM (Song et al., 2020a), DPM-Solver (Lu et al., 2022a) and DPM-Solver++ (Lu et al., 2022b) as the PF-ODE solvers. Proven in (Lu et al., 2022a), the DDIM-Solver is actually the first-order discretization approximation of the DPM-Solver.

For **DDIM** (Song et al., 2020a) , the detailed formula of DDIM PF-ODE solver $\Psi_{\text{DDIM}}$ from $t_{n+k}$ to $t_n$ is provided as follows.

$$
\begin{aligned}
\Psi_{\text{DDIM}}(\boldsymbol{z}_{t_{n+k}}, t_{n+k}, t_n, \boldsymbol{c}) &= \hat{\boldsymbol{z}}_{t_n} - \boldsymbol{z}_{t_{n+k}} \\
&= \underbrace{\frac{\alpha_{t_n}}{\alpha_{t_{n+k}}} \boldsymbol{z}_{t_{n+k}} - \sigma_{t_n} \left( \frac{\sigma_{t_{n+k}} \cdot \alpha_{t_n}}{\alpha_{t_{n+k}} \cdot \sigma_{t_n}} - 1 \right) \hat{\boldsymbol{\epsilon}}_\theta(\boldsymbol{z}_{t_{n+k}}, \boldsymbol{c}, t_{n+k})}_{\text{DDIM Estimated } \boldsymbol{z}_{t_n}} - \boldsymbol{z}_{t_{n+k}}
\end{aligned}
\tag{31}
$$

For **DPM-Solver** (Lu et al., 2022a), we only consider the case for $order = 2$, and the detailed formula of PF-ODE solver $\Psi_{\text{DPM-Solver}}$ is provided as follows. First we define some notations. We denote $\lambda_{t_n} = \log(\frac{\alpha_{t_n}}{\sigma_{t_n}})$, which is the Log-SNR, $h^0_{t_n} = \lambda_{t_n} - \lambda_{t_{n+k}}, h^1_{t_n} = \lambda_{t_n} - \lambda_{t_{n+k/2}}$, and $r_{t_n} = h^1_{t_n}/h^0_{t_n}$.

$$
\begin{aligned}
&\Psi_{\text{DPM-Solver}}(\boldsymbol{z}_{t_{n+k}}, t_{n+k}, t_n, \boldsymbol{c}) \\
&= \frac{\alpha_{t_n}}{\alpha_{t_{n+k}}} \boldsymbol{z}_{t_{n+k}} - \sigma_{t_n}(e^{h^0_{t_n}} - 1)\hat{\boldsymbol{\epsilon}}_\theta(\boldsymbol{z}_{t_{n+k}}, \boldsymbol{c}, t_{n+k}) \\
&\quad - \frac{\sigma_{t_n}}{2r_{t_n}}(e^{h^0_{t_n}} - 1) \left( \hat{\boldsymbol{\epsilon}}_\theta(\boldsymbol{z}^\Psi_{t_{n+k/2}}, \boldsymbol{c}, t_{n+k/2}) - \hat{\boldsymbol{\epsilon}}_\theta(\boldsymbol{z}_{t_{n+k}}, \boldsymbol{c}, t_{n+k}) \right) - \boldsymbol{z}_{t_{n+k}},
\end{aligned}
\tag{32}
$$

where $\hat{\boldsymbol{\epsilon}}$ is the noise prediction model, and $\boldsymbol{z}^\Psi_{t_{n+k/2}}$ is the middle point between $n + k$ and $n$, given by the following formula:

$$
\boldsymbol{z}^\Psi_{t_{n+k/2}} = \frac{\alpha_{t_{n+k/2}}}{\alpha_{t_{n+k}}} \boldsymbol{z}_{t_{n+k}} - \sigma_{t_{n+k/2}}(e^{h^1_{t_n}} - 1)\hat{\boldsymbol{\epsilon}}_\theta(\boldsymbol{z}_{t_{n+k}}, \boldsymbol{c}, t_{n+k})
\tag{33}
$$

For **DPM-Solver++** (Lu et al., 2022b), we consider the case for $order = 2$, DPM-Solver++ replaces the original noise prediction to data prediction (Lu et al., 2022b), with the detailed formula of $\Psi_{\text{DPM-Solver++}}$ provided as follows.

$$
\begin{aligned}
&\Psi_{\text{DPM-Solver++}}(\boldsymbol{z}_{t_{n+k}}, t_{n+k}, t_n, \boldsymbol{c}) \\
&= \frac{\sigma_{t_n}}{\sigma_{t_{n+k}}} \boldsymbol{z}_{t_{n+k}} - \alpha_{t_n}(e^{-h^0_{t_n}} - 1)\hat{\boldsymbol{x}}_\theta(\boldsymbol{z}_{t_{n+k}}, \boldsymbol{c}, t_{n+k}) \\
&\quad - \frac{\alpha_{t_n}}{2r_{t_n}}(e^{-h^0_{t_n}} - 1) \left( \hat{\boldsymbol{x}}_\theta(\boldsymbol{z}^\Psi_{t_{n+k/2}}, \boldsymbol{c}, t_{n+k/2}) - \hat{\boldsymbol{x}}_\theta(\boldsymbol{z}_{t_{n+k}}, \boldsymbol{c}, t_{n+k}) \right) - \boldsymbol{z}_{t_{n+k}},
\end{aligned}
\tag{34}
$$

where $\hat{\boldsymbol{x}}$ is the data prediction model (Lu et al., 2022a) and $\boldsymbol{z}^\Psi_{t_{n+k/2}}$ is the middle point between $n + k$ and $n$, given by the following formula:

$$
\boldsymbol{z}^\Psi_{t_{n+k/2}} = \frac{\sigma_{t_{n+k/2}}}{\sigma_{t_{n+k}}} \boldsymbol{z}_{t_{n+k}} - \alpha_{t_{n+k/2}}(e^{-h^1_{t_n}} - 1)\hat{\boldsymbol{x}}_\theta(\boldsymbol{z}_{t_{n+k}}, \boldsymbol{c}, t_{n+k})
\tag{35}
$$

## F    TRAINING DETAILS OF LATENT CONSISTENCY DISTILLATION

As mentioned in Section 5.1, we conduct our experiments in two resolution settings 512×512 and 768×768. For the former setting, we use the LAION-Aesthetics-6+ (Schuhmann et al., 2022) 12M dataset, consisting of 12M text-image pairs with predicted aesthetics scores higher than 6. For the latter setting, we use the LAIOIN-Aesthetic-6.5+ (Schuhmann et al., 2022), which comprise 650K text-image pairs with predicted aesthetics scores higher than 6.5.

For 512×512 resolution, we train the LCM with the teacher diffusion model Stable-Diffusion-V2.1-Base (SD-V2.1-Base) (Rombach et al., 2022), which is originally trained on 512×512 resolution images using the $\epsilon$-Prediction (Ho et al., 2020). We train LCM (512×512) with 100K iterations

on 8 A100 GPUs, using a batch size of 72, the same learning rate 8e-6 , EMA rate $\mu = 0.999943$ and Rectified Adam optimizer (Liu et al., 2019) used in (Song et al., 2023). We select the DDIM-Solver (Song et al., 2020a) and skipping step $k = 20$ in Eq. 17. We set the guidance scale range $[\omega_{\min}, \omega_{\max}] = [2, 14]$, which is consistent with the setting in Guided-Distill (Meng et al., 2023). During training, we initialize the consistency function $\boldsymbol{f}_{\boldsymbol{\theta}}(\boldsymbol{z}_{t_n}, \omega, \boldsymbol{c}, t_n)$ with the same parameters as the teacher diffusion model (SD-V2.1-Base). To encode the CFG scale $\omega$ into the LCM, we applying Fourier embedding to $\omega$, integrating it into the origin LCM backbone by adding the projected $\omega$-embedding into the original embedding, as done in (Meng et al., 2023). We use a zero parameter initialization method mentioned in (Zhang & Agrawala, 2023) on projected $\omega$-embedding for better training stability. For training LCM ($512 \times 512$), we use a augmented consistency function parameterized in $\epsilon$-prediction as discussed in Appendix. D.

For $768 \times 768$ resolution, we train the LCM with the teacher diffusion model Stable-Diffusion-V2.1 (SD-V2.1) (Rombach et al., 2022), which is originally trained on $768 \times 768$ resolution images using the $\boldsymbol{v}$-Prediction (Salimans & Ho, 2022). We train LCM ($768 \times 768$) with 100K iterations on 8 A100 GPUs using a batch size of 16, while the other hyper-parameters remain the same as in $512 \times 512$ resolution setting.

## G   REPRODUCTION DETAILS OF GUIDED-DISTILL

Guided-Distill (Meng et al., 2023) serves as a major baseline for guided distillation but is not open-sourced. We adhered strictly to the training procedure described in the paper, reproducing the method for accurate comparisons. For $512 \times 512$ resolution setting, Guided-Distill (Meng et al., 2023) used a large batch size of 512, which requires at least 32 A100 GPUs for training. Due to limited resource, we reduced the batch size to 72 (512 resolution), while setting the batchsize to 16 for 768 resolution, the same as ours, and trained for 100K iterations, also the same as in LCM.

Specifically, Guided Distill involves two stages of distillation. For the first stage, it uses a student model to fit the outputs of the pre-trained guided diffusion model using classifier-free guidance scales $\omega$. The loss function is as follows:

$$\mathbb{E}_{w \sim p_w, t \sim \mathcal{U}[0,1], \boldsymbol{x} \sim p_{\text{data}}(\boldsymbol{x})}[\omega(\lambda_t) || \hat{\boldsymbol{x}}_{\boldsymbol{\eta}_1}(\boldsymbol{z}_t, w) - \hat{\boldsymbol{x}}_{\boldsymbol{\theta}}^w(\boldsymbol{z}_t) ||_2^2], \tag{36}$$

where $\hat{\boldsymbol{x}}_{\boldsymbol{\theta}}(\boldsymbol{z}_t) = (1 + w)\hat{\boldsymbol{x}}_{c,\boldsymbol{\theta}}(\boldsymbol{z}_t) - w\hat{\boldsymbol{x}}_{\boldsymbol{\theta}}(\boldsymbol{z}_t), \boldsymbol{z}_t \sim q(\boldsymbol{z}_t|\boldsymbol{x})$ and $p_w(w) = \mathcal{U}[w_{\min}, w_{\max}]$.

In our implementation, we follow the same training procedure in (Meng et al., 2023) except the difference of computation resources. For **first stage distillation**, we train the student model with 25,000 gradient updates (batch size 72), roughly the same computation costs as in (Meng et al., 2023) (3,000 gradient updates, batch size 512), and we reduce the original learning rate $1e-4$ to $5e-5$ for smaller batch size. For **second stage distillation**, we progressively train the student model using the same schedule as in Guided-Distill (Meng et al., 2023) except for batch size difference. We train the student model with 2500 gradient updates except when the sampling step equals to 1,2, or 4, where we train for 20000 gradient updates, using the same schedule as used in (Meng et al., 2023). We trained until the total number of gradient iterations for the entire stage reached 100K the same as in LCM training. The generation results of Guided Distill are shown in Figure 2. We can also see that the performances in Table 1 and Table 2 are similar, further verifying the correctness of our Guided-Distill implementation. Nevertheless, we acknowledge that longer training and more computational resources can lead to better results as reported in (Meng et al., 2023). However, LCM achieves faster convergence and superior results under the same computation cost (same batch size, same number of iterations), demonstrating its practicability and superiority.

## H   COMPARISON OF IMAGE GENERATION METRICS AND INFERENCE TIME

By distilling classifier-free guidance into the model, LCM can generate high-quality images in very short inference time, as shown in Figure 7. We compare the inference time at the setting of 768 x 768 resolution, CFG scale $\omega = 8$, batch size of 4, using an A800 GPU.

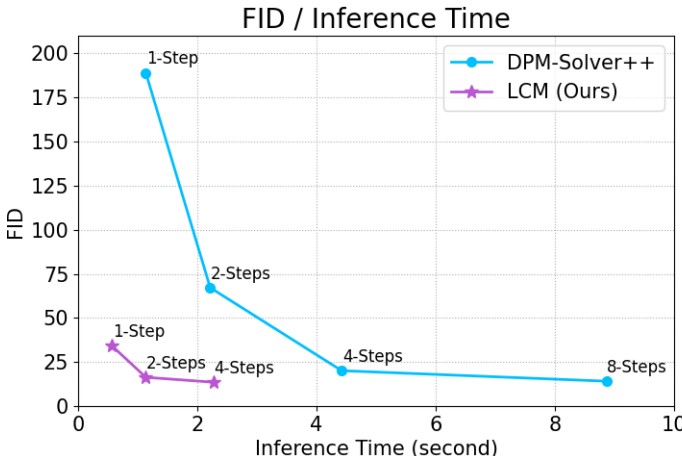

Figure 7: Comparison of FID Scores and Inference Time in Image Generation.

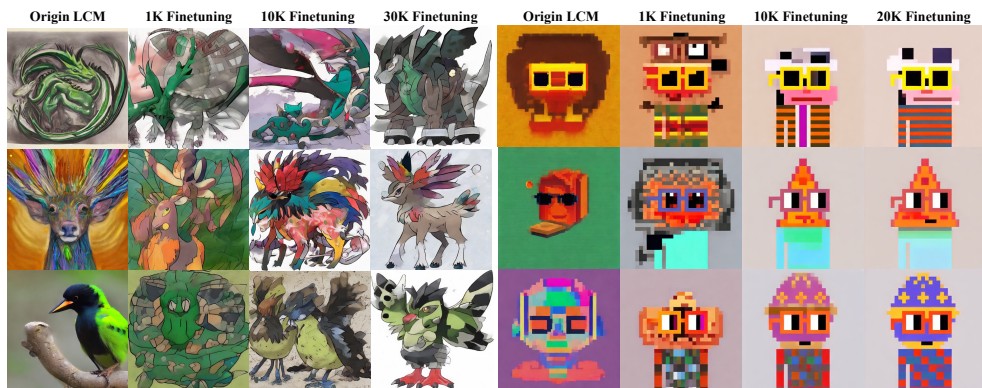

Figure 8: Additional results of 4-step LCMs using Latent Consistency Fine-tuning (LCF) on two customized datasets: Pokemon dataset (left) and Pixel art dataset (right).

## I  MORE LCF FEW-STEP INFERENCE RESULT

In Figure 8, we present further results of LCF. The fine-tuned LCM demonstrates its efficiency by producing images with tailored styles in a small number of steps, underscoring the efficacy of our approach.

## J  MORE FEW-STEP INFERENCE RESULTS

We present more images (768×768) generation results with LCM using 4 and 2-steps inference in Figure 9 and Figure 10. It is evident that LCM is capable of synthesizing high-resolution images with just 2, 4 steps of inference. Moreover, LCM can be derived from any pre-trained Stable Diffusion (SD) (Rombach et al., 2022) in merely 4,000 training steps, equivalent to around 32 A100 GPU Hours, showcasing the effectiveness and superiority of LCM.

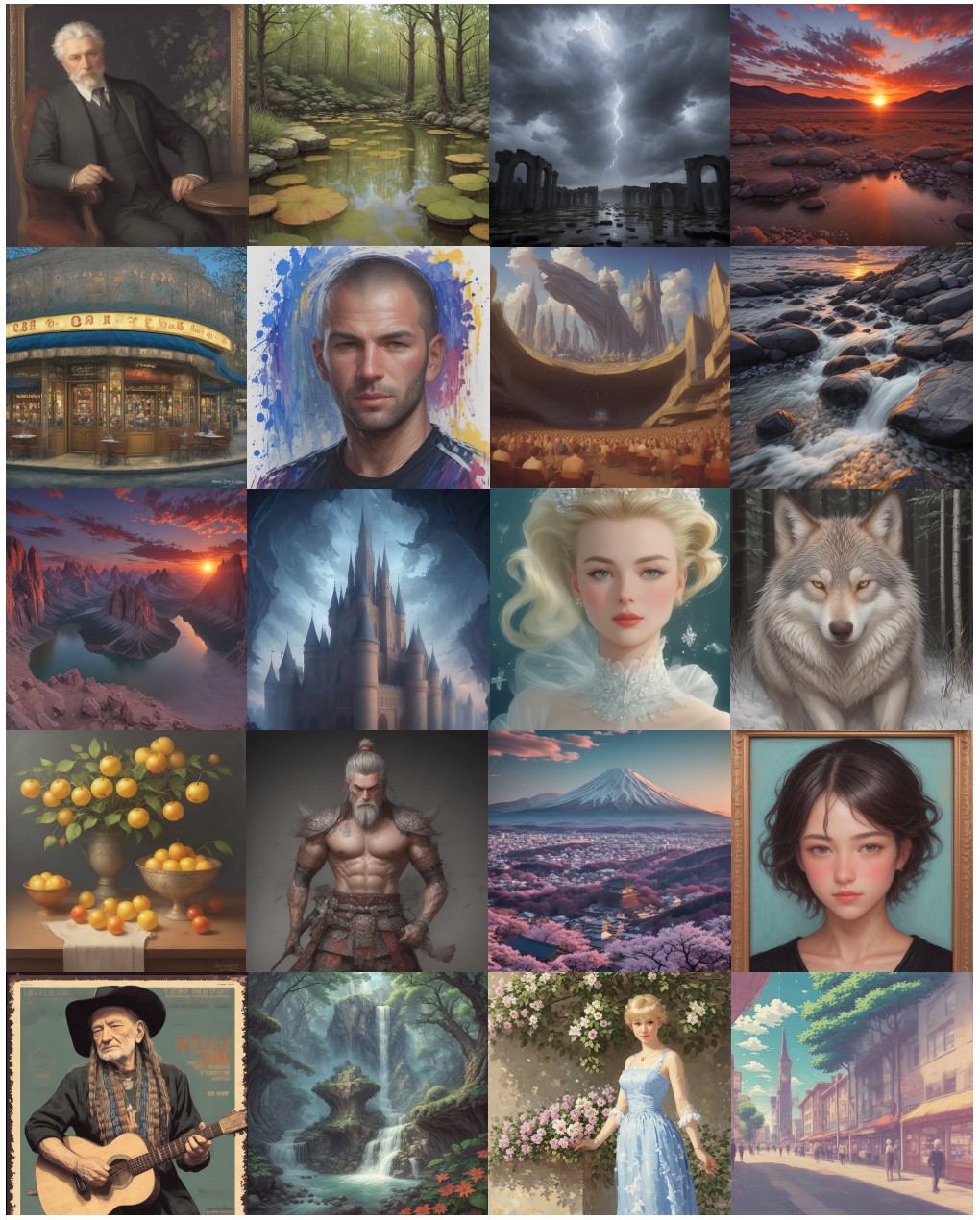

**4-Steps Inference**

Figure 9: More generated images results with LCM 4-Step inference (768×768 Resolution). We employ LCM to distill the Dreamer-V7 version of SD in just 4,000 training iterations.

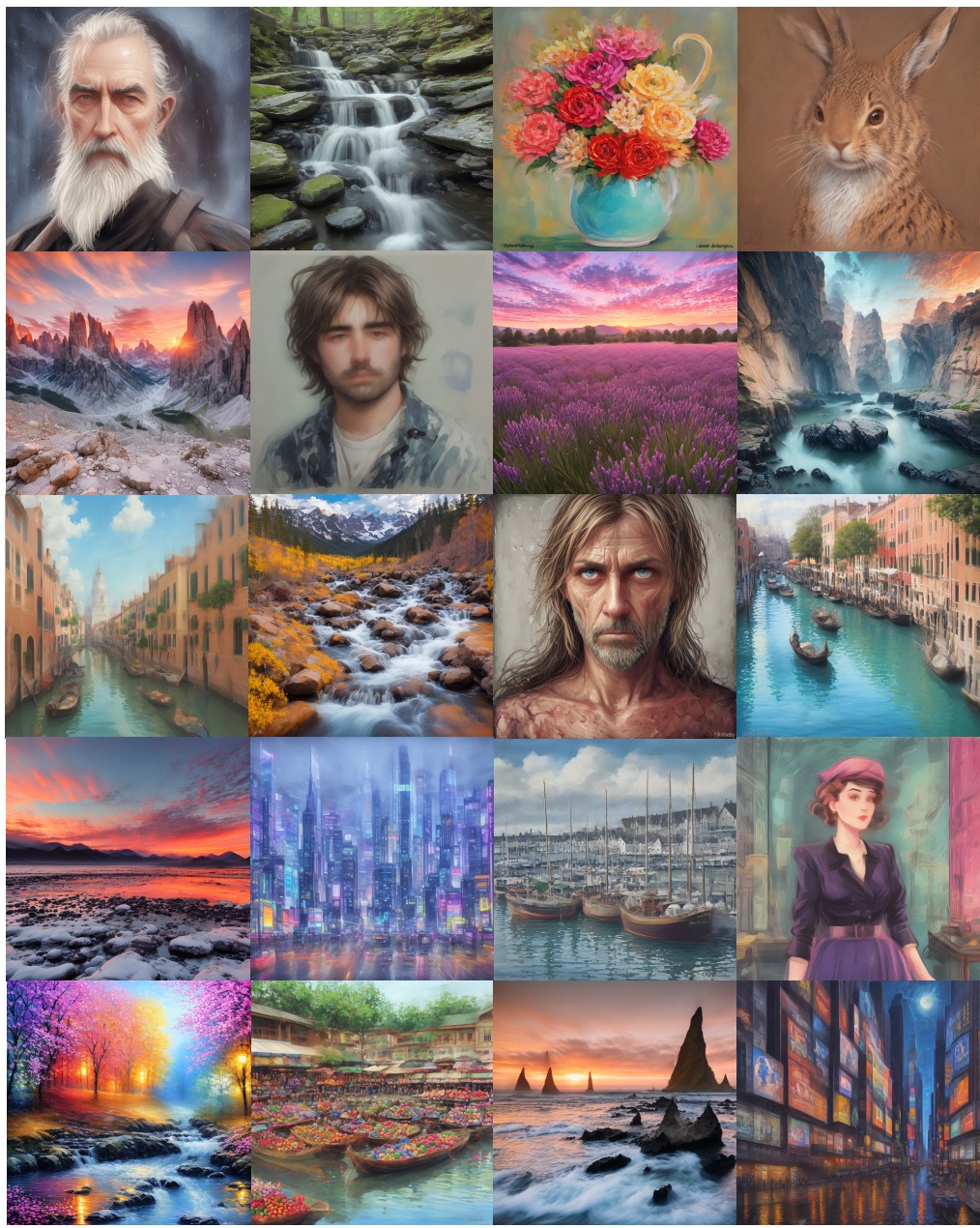

**2-Steps Inference**

Figure 10: More generated images results with LCM 2-steps inference (768×768 Resolution). We employ LCM to distill the Dreamer-V7 version of SD in just 4,000 training iterations.

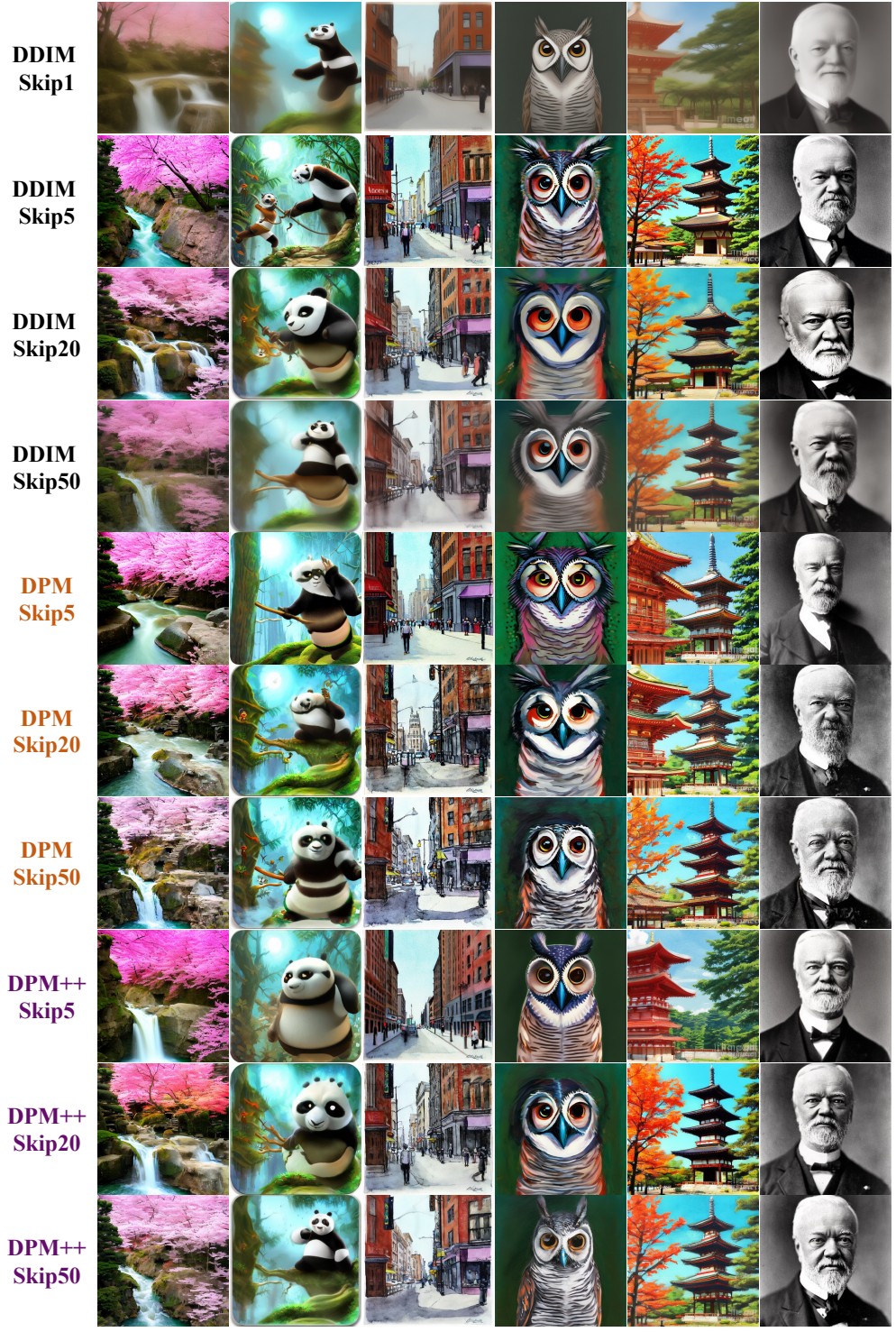

Figure 11: Visualization of using different skipping step schedule $k$ and ODE-Solvers. These models are trained for only 4k iterations (32 A100 GPU hours) and for 4-step inference.

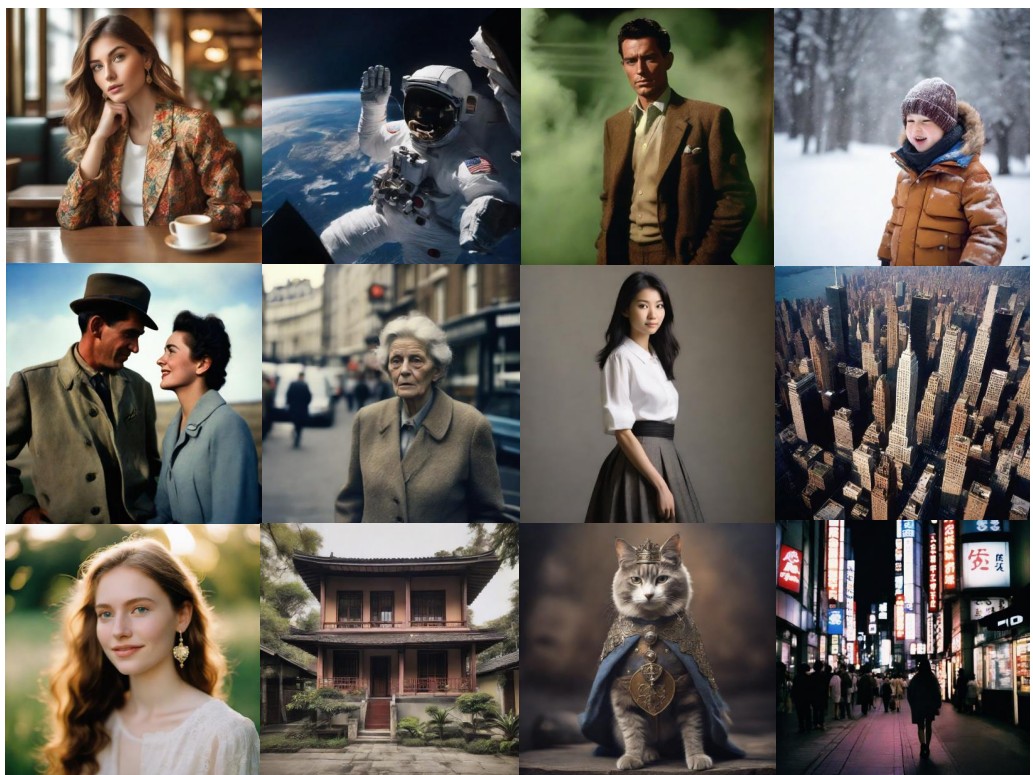

Figure 12: (1024×1024 Resolution) More generated images results with LCM-SDXL 4-Step inference.

## K    MORE VISUALIZATION OF ABLATION STUDY

Figure 11 displays a qualitative comparison of results under different ODE solvers and various skipping steps. It is evident from the figure that the use of skipping steps enhances the quality of the generated images.

## L    MORE FEW-STEP INFERENCE RESULTS WITH LCM-SDXL

We provide additional results of high-resolution image generation using LCM, now at 1024×1024 pixels, as illustrated in Figure 12. This demonstrates that LCM efficiently synthesizes images at higher resolution with a mere 4-step inference process. Furthermore, the adaptability of LCM is emphasized by its capability to evolve from pre-trained SDXL model (Podell et al., 2023) within a reasonably short training period, highlighting its effectiveness and potential.

The prompts for the images are presented below, corresponding to images in Figure 12 in the top-down, left-to-right order.

- a photo of an beautiful young woman wearing a floral patterned blazersitting in cafe, golden lighting, highly detailed, photo realistic.

- High angle photo of an astronaut in space looking at earth –v 5.2 –ar 3:2

- a man in a brown blazer standing in front of smoke, backlit, in the style of gritty hollywood glamour, light brown and emerald, movie still, emphasis on facial expression, robert bevan, violent, dappled –ar 16:9

- photo of a kid playing , snow filling the air –ar 4:3

- rim lighting, a couple looking at each other, color photograph, photograph by Robert Capa –ar 4:3

- analog film photo of old woman on the streets of london . faded film, desaturated, 35mm photo, grainy, vignette, vintage, Kodachrome, Lomography, stained, highly detailed, found footage
- Beautiful asian woman with skirt, Rembrandt Lighting –no painting –ar 16:9
- A bird-eye shot photograph of New York City, shot on Lomography Color Negative 800 –v 5.2 –ar 4:3
- realistic portrait photography of beautiful girl, pale skin, golden earrings, summer golden hour, kodak portra 800, 105 mm fl. 8.
- indone house of indochine style, soft, filter, noise
- (Masterpiece:1. 5), RAW photo, film grain, (best quality:1. 5), (photorealistic), realistic, real picture, intricate details, photo of full body a cute cat in a medieval warrior costume, ((wastelands background)), diamond crown on head, (((dark background)))
- back view of a woman walking at Shibuya Tokyo, shot on Afga Vista 400, night with neon side lighting –v 5.2 –ar 16:9

## M    PROMPTS FOR FIGURE 1

The prompts for the images in Figure 1 (ordered in a top-down, left-to-right manner) are presented below.

**4 steps**

- Jim McVicker, sef Portrait, fine arts, Portraits of Painters
- Massif Torres del Paine Chili - ALEXANDRE DESCHAUMES - Photograph
- Yellow Roses and Peaches, oil, 24 x 24.
- A wolf. Naya's arrival in Belgium completes the return of the predator to every continental country in Europe.
- Motorcyle Digital Art Sunset Artwork
- lesyakostiv-retoucher-lily-red-campaign-11
- Waterfalls Mountains Scenery Crag Fog Landscapes Wallpaper Mural Landscape Wallpaper Landscape Walls Paint By Number
- Painting © by Tamara Natalie Madden African American art
- Pink Landscape Wallpapers Top Free Pink Landscape Backgrounds Wallpaperaccess
- "Art Print Exclusive Serie - Ferrari 275GTB ""River Side"" - Artist: Keith Woodcock"
- oil portrait of a business man sitting
- Sunrise-Tamarama Beach Sydney Australia Painting by Chris Hobel

**2 steps**

- Sunset in the Tre Cime di Lavaredo, Dolomites, Italy, Europe
- Leah (pastel) by David Wells
- painting-of-forest-and-pond
- The Lights of the City - Cross Stitch Chart - Click Image to Close

**1 step**

- by Stephen Barker - Landscapes Waterscapes (lake district, Derwent, island, early morning, long exposure)
- Fall And Autumn Wallpaper Daniel Wall Rainy Day In Autumn Painting Oil Artwork
- Painting - Magical Night In New York by Chin H Shin
- Jim McVicker, self Portrait, fine arts, Portraits of Painters

| LCM 4-Step (Ours) | SDXL 4-Step | SDXL 8-Step | SDXL 16-Step | SDXL 25-Step |
| --- | --- | --- | --- | --- |

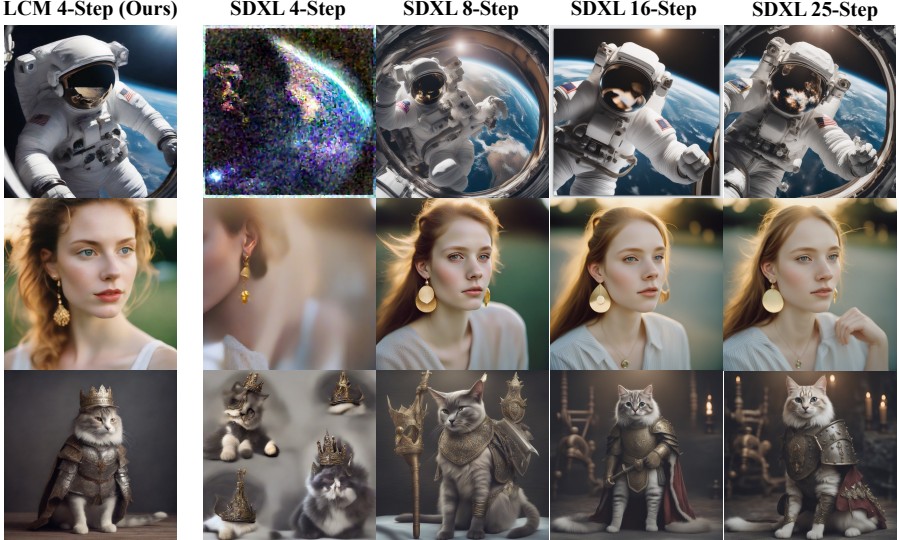

Figure 13: (1024×1024 Resolution) More generated images comparison results with LCM-SDXL 4-Step inference.

## N  LCM SD-XL COMPARSION

We include a detailed comparison of image qualities produced by LCM-SDXL and SDXL with DPM Solver across various sampling stages in Figure 13. Notably, LCM-SDXL exhibits a significant efficiency, achieving image quality on par with the original SDXL's 25-step process with DPM Solver in merely 4 steps, highlighting its substantial superiority.

