# OpenReview forum: "Latent Consistency Models: Synthesizing High-Resolution Images with Few-step Inference"
_ICLR.cc/2024/Conference — Submitted to ICLR 2024_

### Official Review · Reviewer_SB2d · 2023-10-30

**Soundness:** 3 good
**Presentation:** 3 good
**Contribution:** 1 poor
**Rating:** 5
**Confidence:** 4

**Summary:**

This paper proposes to train consistency models using a pretrained large latent diffusion model (i.e., Stable Diffusion). The main difference between the original consistency models follows: (a) Consideration of augmented ODE with guidance scale $\omega$ to enable the distillation in a single stage, (b) use skipping timesteps for distillation since pretrained Stable Diffusion uses larger timestep (1,000) than original consistency models that use EDM formulation.

**Strengths:**

- The paper is generally well-written and easy to follow.
- Compared with other efficient sampler or distillation methods, the proposed method shows a considerable performance improvement in a smaller sampling step regime.

**Weaknesses:**

- My major concern is about the technical novelty and contribution of this work. The paper argues there are two main contributions: (a) usage of augmented ODE and (b) skipped timestep (e.g., $k=20$) for better distillation. For (a), it seems quite straightforward to consider augmented ODE since the Stable Diffusion uses cfg. For (b), it also seems straightforward since Stable Diffusion uses a large timestep ($T=1000$), unlike EDM formulation in the original consistency model paper, and thus, one can easily expect using consecutive timesteps for distillation is inefficient. In these respects, I think both (a) and (b) looks too straightforward and marginal technical contribution to be accepted at ICLR.
-  Title -- "Latent" Consistency model? I expected the authors to consider unique aspects of the "latent" diffusion model in incorporating the concept of consistency models; for instance, the original consistency model paper argues using perceptual metrics is efficient for better distillation. However, it seems the proposed method does not depend on whether the pretrained model is a latent diffusion model or not; it can be applied to any diffusion model that uses a large timestep (e.g., $T=1000$); thus, I think including "latent" in the title is unnecessary.
- For figures for qualitative illustrations, it's better to provide text prompts to show the image-text alignment as well.

**Questions:**

- Why is the DDIM sampler mainly used for LCD (main experiments; Table 1 and 2)? What if the method uses a different ODE solver/sampler?
- Which metric does LCD use for distillation? The original consistency model paper mainly uses the LPIPS score, but it seems such a metric is not applicable since this method deals with latent diffusion models.

---

> ### Author Response · Authors · 2023-11-19
> **Response to Reviewer SB2d**
>
> We address your concerns below.
>
> ### 1. About technical novelty and contribution of this work.
>
> We understand the concerns raised about the perceived straightforwardness of our contribution. Still, we politely ask for a reevaluation of our paper contribution. To clarify:
>
> 1) A key difference between the original consistency models [1] and LCM lies in the transformation of the **empirical PF-ODE** (Eq.24) from the original consistency model to a **more general PF-ODE** (Eq.8). This was not addressed in the original model [1]. We also adapted the Euler/Heun solver under EDM schedule [2] to a more compatible solver for Stable Diffusion (see Eq.18 and Appendix E, including DDIM, DPM, DPM++ solvers).  Our approach expands the original model's $x$-prediction (Eq.24) to include $\epsilon$-prediction and $v$-prediction for Stable Diffusion, **offering broader parameterizations as shown in Eq. 9 and Appendix D**. These transformations and adaptations are **non-trivial and are critical** for LCM's success. However, we have weakened these difference in the main paper for readers much easier to read.
>
> 2) Our method successfully combines latent consistency distillation (LCD), one-stage guided consistency distillation (CFG), and skipping-step techniques to achieve a highly efficient **single-stage** guided distillation for text-to-image tasks. This represents a significant advancement over previous two-stage methods like Guided-Distill [3]. With only 32 A100 GPU hours, we achieve impressive 2~4-step inference results, **demonstrating both efficiency and effectiveness**.
>
> **Simplicity and effectiveness**, far from being limitations, are vital principles in AI research. Our work not only identifies critical components for effective LDM distillation but also achieves remarkable few-step inference results, paving the way for real-time generative AI applications. **We strongly require a reevaluation of our paper, and believe firmly that LCM have made enough contribution to this field**.
>
> ### 2. About title.
> We thank the reviewer for their attention to the choice of terminology in our paper’s title. The title ``Latent Consistency Models" was chosen deliberately to acknowledge two prior seminal papers titled “Latent Diffusion Models” (Rombach et al.) and “Consistency Models” (Song et al.) and we believe our title is an accurate reflection of the essence of our work. The choice to include “latent” in the title emphasizes the context in which the consistency model is applied. In particular, our research is primarily based on the most widely used open-sourced latent diffusion models (i.e., Stable Diffusion). We utilize the same autoencoder framework as these models, which is central to our method. Furthermore, our experiments demonstrate that our models not only produce generation quality comparable to existing latent diffusion models but also significantly expedite the sampling process. While it is true that some of the methods we proposed (e.g., the skipping-step technique) could be adapted to other diffusion models, the effectiveness and efficiency of our approach are particularly pronounced in the latent space, as demonstrated by our experiments. Moreover, we think the inclusion of the word “latent” helps delineate our work from the original Consistency Model by Song et al, thereby avoiding potential confusion. In conclusion, we think our title is carefully chosen to honor prior works while accurately representing the core innovation of our work. We hope this clarifies our rationale and justifies the choice of our title’s wording.
>
> ### 3. About text prompts
>
> Thanks for pointing out. We have updated the main paper. The text prompts of Figure 1 are provided in **Appendix.M**. Also we provide the LCM SD-XL [4] results in **Appendix.L (Figure.12)**.
>
> ### 4. About experiments
>
> > 4.1. Why is the DDIM sampler mainly used for LCD for main experiments? What if the method uses a different ODE solver?
>
> In Figure 3, We present the ablation results for different ODE solvers and the number of skipping steps. It is evident that under our chosen number of skipping steps (20 steps), DDIM, DPM-Solver, and DPM-Solver++ demonstrate similar experimental outcomes. As a result, we selected DDIM to complete our main experiments. We believe that the experimental results using DDIM as the ODE solver are sufficiently representative and effective in terms of the performance metrics achieved by our method.
>
> > 4.2 Which metric does LCD use for distillation.
>
> Thank you for pointing out this issue. Since we have already employed an autoencoder to encode the original images into the latent space, the LPIPS metric used in the original Consistency Models paper [1] is not appropriate in the latent space. We used the L2 distance in the latent space as our distance metric. We have updated our paper to make this information more explicitly.

---

> > ### Author Response · Authors · 2023-11-19
> > **Response to Reviewer SB2d**
> >
> > Reference:
> >
> > [1] Song, Yang, et al. "Consistency models." (2023).
> >
> > [2] Karras, et al. "Elucidating the design space of diffusion-based generative models" (2022)
> >
> > [3] Meng, et al. "On distillation of guided diffusion models" (2023)
> >
> > [4] Podell, Dustin, et al. ”Sdxl: Improving latent diffusion models for high-resolution image synthesis.” arXiv preprint arXiv:2307.01952 (2023).

---

> ### Author Response · Authors · 2023-11-22
> **Reminder to Reviewer SB2d**
>
> Dear Reviewer,
>
> Thank you for reviewing our paper. We hope our responses have addressed your concerns. If so, we kindly ask if you could reconsider your score for our work.
>
> Should you have any further questions, we're ready to discuss and clarify.
>
> Many thanks for your time and effort.

---

> ### Author Response · Authors · 2023-11-22
> **Deadline coming. looking forward to your feedback.**
>
> Dear Reviewer SB2d, We kindly request your feedback as the rebuttal deadline is approaching in less than 1 days. We hope our previous feedback addresses your concern. We would like to thank you again for your time and previous review and we are looking forward to further discussions.

---

### Official Review · Reviewer_pn9B · 2023-11-01

**Soundness:** 3 good
**Presentation:** 2 fair
**Contribution:** 3 good
**Rating:** 6
**Confidence:** 3

**Summary:**

This paper proposes latent consistency models for fast high-resolution image generation. In addition, it provides a simple and efficient one-stage guided consistency distillation method for few-step (2∼4) or even 1-step sampling. Experiments show that the LCMs achieves state-of-the-art text-to-image generation performance with few-step inference.

**Strengths:**

1. The idea is novel and interesting. The author proposes latent consistency models that leverage consistency model in latent space, achieving few-step or even one-step sampling.
2. The experimental results look impressive. The latent consistency model outperforms state-of-the-art methods by large margin especially with one step.

**Weaknesses:**

1. The paper is not well-organized. The introduction of the proposed method in Sec.1 is too concise. The description and motivation for each design should be more detailed. Figure.1 takes up too much space and can be reduced appropriately.
2. Insufficient content for related work.
3. Lack of ablation studies. The authors should provide qualitative results of the ablation studies on the ODE solvers & skipping-step schedule as well as qualitative and quantitative results on guided consistency distillation.

**Questions:**

1. What's the overall pipeline of the proposed method? It seems that the authors do not describe or show the pipeline in detail.
2.  It is better to show the comparisons between training time and memory.

---

> ### Author Response · Authors · 2023-11-19
> **Response to Reviewer pn9B**
>
> We thank the constructive feedback from you. We address your concerns below.
>
> ### 1. The paper is not well-organized. .... Figure 1. takes up too much space.
>
> Thanks for pointing out. We have updated the paper. Adding more content to the intro. And we add more details to the motivation, reducing the Figure.1 space.
>
> ### 2. Insufficient content for related work.
>
> Thanks for pointing out. We have added more related work content in the updated paper.
>
> ### 3. Lack of ablation studies.
>
> > 3.1 Qualitative comparison of different ODE-solvers and skipping steps
>
> Thanks. **Figure 3 clearly demonstrates the effects of using different ODE-solvers and skipping-step schedules**, making further visualization in the main paper somewhat redundant. Nonetheless, we've included updated qualitative ablation study results on these aspects in **Appendix.K (Figure 11)**, supporting conclusions similar to those in Section 5.2 and Figure 3, particularly the importance of step-skipping for faster training and the drawbacks of excessively large skipping steps.
>
> > 3.2 Qualitative and quantitative results on guided consistency distillation.
>
> Thanks. Our qualitative and quantitative results on guided consistency distillation **are already presented in Figures 4 and 5**. It is important to note that using a very small Classifier-free guidance (CFG) scale is essentially equivalent to disabling CFG. From Figure 4, it is evident that a smaller CFG scale ($\omega=2$) results in a lower CLIP Score compared to a larger scale. Moreover, an increased CFG scale significantly enhances image quality, as shown by the increasing CLIP Score in Figure 4 and the visualization results in Figure 5. In conclusion, our experimental results have clearly demonstrated the effectiveness of one-stage guided consistency distillation.
>
> ### 4. About overall pipeline of LCM.
>
> Thanks for asking. LCM's pipeline, mainly outlined in Algorithm 1, consists of three main steps: (1) Start with any pretrained LDM; (2) Apply the latent consistency distillation (LCD) method from Algorithm 1 to convert the LDM into an LCM; (3) Employ the multi-step inference method in Algorithm 3 for fast sampling from the LCM.
>
> ### 5. better to show the comparisons between training time and memory.
>
> Thank you for your suggestion. In terms of training time, both Guided Distill [1] and our LCM method are almost the same, each trained for the same number of iterations (100K, as detailed in Section 5.1) and use same batch sizes. Regarding memory usage, both methods exhibit similar requirements, with Guided Distill consuming approximately 71.1GB and LCM also using 71.3GB of memory for global batchsize=$72$ with 8 A100s. We would like to mention that LCM has potential for further training optimizations, such as mixed precision training and gradient checkpointing techniques. These considerations are beyond the current scope of our paper and left as future improvement.
>
> References:
>
> [1] Meng, et al. "On distillation of guided diffusion models" (2023)

---

> ### Author Response · Authors · 2023-11-22
> **Reminder to Reviewer pn9B**
>
> Dear Reviewer,
>
> Thank you for reviewing our paper. We hope our responses have addressed your concerns. If so, we kindly ask if you could reconsider your score for our work.
>
> Should you have any further questions, we're ready to discuss and clarify.
>
> Many thanks for your time and effort.

---

> ### Author Response · Authors · 2023-11-22
> **Deadline coming. looking forward to your feedback.**
>
> Dear Reviewer pn9B, We kindly request your feedback as the rebuttal deadline is approaching in less than 1 days. We hope our previous feedback addresses your concern. We would like to thank you again for your time and previous review and we are looking forward to further discussions.

---

### Official Review · Reviewer_24Xw · 2023-11-01

**Soundness:** 3 good
**Presentation:** 3 good
**Contribution:** 3 good
**Rating:** 6
**Confidence:** 4

**Summary:**

In this paper, the authors apply the consistency model to the latent diffusion, significantly reducing the inference steps in diffusion models. They also implement guided distillation, enhancing quality through classifier-free guidance, and introduce time step skipping to expedite the distillation process. The effectiveness is demonstrated through experiments on LAION subsets.

**Strengths:**

* The paper is well-written and the method is intuitive to understand.
* The results are impressive. The proposed method can significantly reduce the sampling steps of the diffusion models while achieving a decent quality performance.

**Weaknesses:**

* The authors should benchmark their approach against the single-step diffusion model [InstaFlow](https://github.com/gnobitab/InstaFlow) [1], and also include results from the original 50-step Stable Diffusion as a baseline. It's currently unclear how their method's speed gains affect performance.
* The proposed latent consistency fine-tuning seems not working, as shown in Figure 6. On the Simpsons dataset, the quality of 30K finetuning is worse than the original LCM.
* The paper lacks results on realistic photo generation, featuring only artistic illustrations. Including realistic photo results would strengthen the evaluation.
* The paper's novelty appears limited. It primarily adapts consistency models to latent space and uses guided distillation in Meng et al. [2] to support Classifier-Free Guidance. While skipping time steps shows efficacy, it doesn't substantially elevate the paper's technical novelty. Are there some challenges of applying consistency models to latent space compared to pixel space?

[1] InstaFlow: One Step is Enough for High-Quality Diffusion-Based Text-to-Image Generation

[2] On Distillation of Guided Diffusion Models

**Questions:**

* No corresponding text prompts in the visual results (e.g., Figure 1).
* Typo. Section 3 Preliminaries -- Diffusion Models: "origin data distribution" -> "original data distribution"

---

> ### Public Comment · ~Hongjian_Liu1 · 2023-11-15
> **Evaluating LCF on large datasets**
>
> It seems that LCF does not work well in Figure 6. Can you evaluate LCF on a larger dataset like CoCo and provide the loss curve,  metrics, and visual results?

---

> ### Author Response · Authors · 2023-11-19
> **Response to Reviewer 24Xw**
>
> We sincerely thank the constructive feedback from you. We address your concerns below.
>
> ### 1. About benchmarking.
>
> > 1.1 should benchmark their approach against the single-step diffusion model InstaFlow.
>
> Thank you for pointing out the InstaFlow paper. We acknowledge the significance of InstaFlow [1], which was released on September 12th, as indicated at https://arxiv.org/abs/2309.06380, just two weeks prior to the ICLR submission deadline. We became aware of this paper even a few days later. Due to such a short timeframe and the absence of publicly available code and pretrained models for InstaFlow, reproducing their work for a detailed comparison was not feasible within our project timeline.
> Additionally, as per the guidelines provided in the ICLR FAQ section (https://iclr.cc/Conferences/2024/ReviewerGuide#FAQ), recent works published **within four months of the submission deadline are not required as baseline algorithms for comparison**. Given that InstaFlow appears to be submitted to the same ICLR24 conference, we consider it appropriate to regard it as concurrent work, aligning with standard academic practices.
> We appreciate your understanding in this matter and are open to future comparisons with InstaFlow as part of our ongoing research, should the necessary resources become available.
>
> > 1.2 Include results from the original 50-step Stable Diffusion as a baseline.
>
> Thanks for your suggestion. We include additional **50-Step DDIM Sampler** Stable Diffusion results here and we have also added it in the caption of Table 1 and 2.
>
> (512$\times$512 Resolution) FID: 10.74 ,  CLIP-Score: 30.34
>
> (768$\times$768 Resolution) FID: 12.74 ,  CLIP-Score: 30.82
>
> > 1.3: Aboust speed gains comparison
>
> Thanks for your suggestion. We also include the inference time in **Appendix.H (Figure.7)** for your reference. It can be noticed that LCM achieve at least **$4\times$ acceleration** copmared with DPM-Solver++.
>
> ### 2. About latent consistency fine-tuning.
>
> We appreciate your interest in our approach to latent consistency fine-tuning (LCF). Upon reflection, we recognize that the Simpsons dataset may not have been the optimal choice for demonstrating fine-tuning capabilities. This is primarily because the original Stable Diffusion (SD) model already exhibits a strong proficiency in generating images in the distinctive style characteristic of the Simpsons. To provide a more comprehensive evaluation, we have supplemented our fine-tuning results **with two additional datasets**, the details of which can be found in **Section 5.3 (Figure.6)** and **Appendix.I (Figure.8)**. Moreover, it's important to highlight that our primary contribution lies in introducing LCF as a **viable alternative for fine-tuning LCM directly** instead of teacher diffusion model, showcasing its potential effectiveness. While there is certainly room for further refinement and improvements in LCF, we believe that these aspects represent exciting avenues for future research. Our current work lays the groundwork for such explorations.
>
> ### 3. Lacks results of realistic photo generation.
>
> Thank you for your valuable observation regarding the lack of results for realistic photo generation in our paper. We agree that this aspect is important for a comprehensive evaluation of our approach.
>
> To clarify, the style of generative images in our study is significantly influenced by the choice of the teacher diffusion model. In Figure 1, we employed Dreamshaper-V7, a popular variant of Stable Diffusion that is fine-tuned specifically for artistic styles. Additionally, our testing utilized text prompts from LAION-5B-Aesthetic6.5+, a dataset that predominantly features aesthetically oriented prompts, resulting in a natural inclination toward artistic style generation in our results.
>
> We acknowledge that Dreamshaper-V7, being fine-tuned on artistic images, is not the ideal model for generating realistic photos. To address this, we have included results of LCM applied to **SD-XL** [2], which represents a more challenging LDM, to demonstrate our method's capability in realistic image generation. These photo-realistic generation results with 4-step inference, underscore the effectiveness of LCM. You can find these additional results in **Appendix.L (Figure.12)**. We hope this addition adequately addresses your concern and highlights the broader applicability of LCM, including in scenarios requiring high-quality, photo realistic image generation.

---

> > ### Author Response · Authors · 2023-11-19
> > **Response to Reviewer 24Xw**
> >
> > ### 4. About paper novelty.
> > > 4.1. It primarily adapts consistency models ... elevate paper's technical novelty.
> >
> > We understand the concerns raised about the perceived straightforwardness of our contributions. We acknowledge (in fact, happily) the conceptual simplicity of our work and **firmly believe that such simplicity is a strength, rather than a shortcoming**, in this important line of research with a vast and fast-growing literature and a multitude of possible technical paths that may be taken to accelerate inference. A simple and effective method offers clear advantages over more complex ones, including easier implementation, optimization and deployment. In fact, during our research, we explored several more complex techniques and methods. However, those seemingly more “fancy” approaches either yield only marginal performance gains or require substantial training (hence computationally prohibitive). Therefore, we opted for simplicity, choosing to exclude those methods in favor of a direct, efficient, and effective approach. Despite the simplicity, we believe LCM offers clear and significant contributions. **LCM is a highly efficient method that first achieves single-stage guided distillation on high resolution text-to-image generation, improving prior work by a large margin in the 2-4 step regime.** Also perhaps because of its simplicity and effectiveness, LCM has already been adopted extensively in a variety of AIGC contexts by the open-sourced community, enabling users to achieve near real-time generations (Due to the ICLR anonymity policy, we do not provide the link here).
> > **In conclusion, we argue that the conceptually simplicity of our contributions should not be seen as a lack of novelty, but rather a virtue.**
> >
> > > 4.2. Are there some challenges of applying ... compared to pixel space.
> >
> > Thanks for your question. Although our approach is conceptually simple, we argue that it is **non-trivial**, especially given a large volume of recent works and many methods that have been proposed to accelerate inference. We have also explored
> > several more complex (and seemingly more novel) techniques and methods but those yield only marginal performance gains. Nevertheless, let us highlight some of the technical challenges:
> >
> > (1). We **transform** the empirical PF-ODE (as per Equation 24) and its corresponding solver (as per Equation 7) in the original consistency model paper [1] to a **more general form** that can be made compatible with Stable Diffusion (PF-ODE, as illustrated in Equation 8, and its corresponding solver, as in Equation 18). We also **reformulate the parameterization** shown in Eq. 9 and Appendix D. These transformation and reformulation are **non-trivial and a crucial factor** for the successful implementation of LCM.
> >
> > (2). The introduction of latent consistency distillation (LCD), single-stage guided distillation (Classifier-Free Guidance or CFG), and the skipping-step techniques are also **critical elements for the success of LCM**. The single-stage guided distillation has a significant impact for improving image quality (see Figure 4, 5). While the skipping-step techniques accelerate the convergence for at least $6\times$, which is shown in Figure.3.
> >
> > (3). Last but not least, we would like to mention that the engineering and system endeavors are also nontrivial. The task of integration latent consistency models and Stable Diffusion required in-depth understanding and engineering with the Stable Diffusion framework. Our careful engineering has results in a robust system that not only achieves state-of-the-art performance in research, while also delivering the practical effectiveness immediately ready for real-world applications.
> >
> > In sum, we had to overcome several technical challenges to deliver the impressive experimental results and efficiency achieved by LCM. **We believe that our efforts are nontrivial and our contributions are significant and noteworthy.**
> >
> >
> > ### 5. Other Questions.
> >
> > > 5.1. No corresponding text prompts in the visual results (e.g., Figure 1)
> >
> > Thanks for pointing out. We have updated the main paper. The text prompts of Figure 1 are provided in **Appendix.M**. Also we provide the LCM SD-XL [2] results in **Appendix.L (Figure.12)**.
> >
> > > 5.2. Typo
> >
> > Thanks. We have fixed the typo in the updated main paper.
> >
> > References:
> >
> > [1] Liu, Xingchao, et al. "Instaflow: One step is enough for high-quality diffusion-based text-to-image generation." arXiv preprint arXiv:2309.06380 (2023).
> >
> > [2] Podell, Dustin, et al. "Sdxl: Improving latent diffusion models for high-resolution image synthesis." arXiv preprint arXiv:2307.01952 (2023).

---

> ### Author Response · Authors · 2023-11-22
> **Reminder to Reviewer 24Xw**
>
> Dear Reviewer,
>
> Thank you for reviewing our paper. We hope our responses have addressed your concerns. If so, we kindly ask if you could reconsider your score for our work.
>
> Should you have any further questions, we're ready to discuss and clarify.
>
> Many thanks for your time and effort.

---

> ### Author Response · Authors · 2023-11-22
> **Deadline coming. looking forward to your feedback.**
>
> Dear Reviewer 24Xw, We kindly request your feedback as the rebuttal deadline is approaching in less than 1 days. We hope our previous feedback addresses your concern. We would like to thank you again for your time and previous review and we are looking forward to further discussions.

---

> > ### Comment · Reviewer_24Xw · 2023-11-23
> > **Official Comment from Reviewer 24Xw**
> >
> > Thank the authors for their response. Some of my concerns have been addressed, but I still have several questions:
> >
> > - Regarding Tables 1 and 2, could you elaborate on the experimental setup, particularly how you split the dataset into training and validation sets? It seems that you train LCM and evaluate the model on the same dataset. I am not sure if Guided-Diffusion follows the same protocol for evaluation and if the comparisons are fair.
> > - For the COCO results, the photorealism in the generated images seems not that high. It would be nice if the authors could include the original SDXL results and also provide some quantitative evaluation on the COCO captions.
> > - Equation 8 in your paper seems to come from Song et al. [1] and Lu et al. [2] as mentioned. What is the relation between it and Equation 24 or 7 in Consistency Model?
> >
> > [1] Score-based generative modeling through stochastic differential equations.
> >
> > [2] Dpm-solver: A fast ode solver for diffusion probabilistic model sampling in around 10 steps.

---

> ### Author Response · Authors · 2023-11-23
> **Response to Reviewer 24Xw**
>
> Thanks for your feedback. We address your concerns below.
>
> > Regarding Tables 1 and 2, could you elaborate on the experimental setup, particularly how you split the dataset into training and validation sets? It seems that you train LCM and evaluate the model on the same dataset. I am not sure if Guided-Diffusion follows the same protocol for evaluation and if the comparisons are fair.
>
> We conduct LCM experiment on Table 1 and Table 2 respectively. For Table 1, we use LAION-5B-Aesthetic-6.0+ dataset, containing 12M image-text pairs, and we train the LCM with $512\times512$ on this dataset. For Table 2, we use the LAION-5B-Aesthetic-6.5+ dataset, containing 650M image-text pairs, and we train the LCM with $768\times768$ resolution. For both dataset, we use 95\% of the data set as the training set and the remaining 5\% as the test set. We use 10,000 prompts from the test set for evaluation. The Guided-Distill follows the same protocal. We trained the Guided-Distill on the same dataset and use the same prompt for evaluation. The total training iteration and batch size is the same as LCM. The comparsions are totally fair and can reflect our LCM superiority.
>
> > For the COCO results, the photorealism in the generated images seems not that high. It would be nice if the authors could include the original SDXL results and also provide some quantitative evaluation on the COCO captions.
>
> In Appendix N, we provide supplemental information detailing the qualitative results of images produced by both LCM-SDXL and SDXL with DPM Solver across various sampling steps. We can observe that LCM-SDXL is capable of generating images of comparable quality to that of the original SDXL's 25-step output with DPM Solver, in merely 4 steps, thus indicating a significant acceleration. However, due to the extensive overhead involved in the training process, we were unable to provide experimental results on the COCO dataset within such a short timeframe. On the other hand, LAION-5B is a more challenging dataset compared to COCO and the experimental results obtained from the LAION-5B dataset have demonstrated the effectiveness of our method.
>
> > Equation 8 in your paper seems to come from Song et al. [1] and Lu et al. [2] as mentioned. What is the relation between it and Equation 24 or 7 in Consistency Model?
>
> Equation (8) presents the general form of the PF-ODE, where the specific expressions for $f(t)$ and $g(t)$ in the equation vary depending on the noise schedule. When the EDM schedule [3] is chosen as done in the original Consistency Models by Song et al. [4], Equation (8) simplifies to Equation (24). By a straightforward discretization, we can get Equation (7) (the same as Equation (25).)
>
> [1] Score-based generative modeling through stochastic differential equations.
>
> [2] Dpm-solver: A fast ode solver for diffusion probabilistic model sampling in around 10 steps.
>
> [3] Karras, Tero, et al. "Elucidating the design space of diffusion-based generative models." Advances in Neural Information Processing Systems 35 (2022): 26565-26577.
>
> [4] Song, Yang, et al. "Consistency models." (2023).

---

### Official Review · Reviewer_dEPF · 2023-11-02

**Soundness:** 3 good
**Presentation:** 3 good
**Contribution:** 2 fair
**Rating:** 5
**Confidence:** 3

**Summary:**

This paper proposes latent consistency models (LCMs) to swiftly inference with minimal steps on any pre-trained LDMs, e.g., Stable Diffusion. LCMs are designed to directly predict the solution of an augmented probability flow ODE in latent space to allow rapid, high-fidelity sampling. A latent consistency fine-tuning (LCF) is further introduced to fine-tune LCMs on customized image datasets. Experiments on the LAION-5B-Aesthetics dataset demonstrates the effectiveness of the proposed LCMs.

**Strengths:**

+ The idea is interesting to view the guided reverse diffusion process as solving an augmented probability flow ODE.

+ The performance looks good on both qualitative and quantitative results and in some cases the results of the proposed method with less steps are better than those of other methods with more steps.

+ Some ablation studies are provided to facilitate the understanding of how the performance benefits from different components, including ODE solvers, skipping-step schedule and guidance scale.

**Weaknesses:**

- Although the authors claim several contributions, it is not clear which ones have the most significant impact on efficiency and quality.

- What is the computational complexity of solving augmented PF-ODE?

- The experiments shows that the proposed method reduces the inference steps, however, how much faster is the inference time exactly compared with other methods?

- What about the performance when the proposed LCMs are applied to other LDMs besides Stable Diffusion?

**Questions:**

1. It is not clear which contributions have the most significant impact on efficiency and quality.

2. What is the computational complexity of solving augmented PF-ODE?

3. How much faster is the inference time exactly compared with other methods?

4. What about the performance when the proposed LCMs are applied to other LDMs besides Stable Diffusion?

---

> ### Author Response · Authors · 2023-11-19
> **Response to Reviewer dEPF**
>
> We sincerely thank the constructive feedback from you. We address your concerns below.
>
> > 1. Not clear about which ones have the most significant impact on efficiency and quality.
>
> Thanks for your question. Our skipping-step technique is the most critical component for speeding up the convergence (i.e., efficiency) and the one-stage guided consistency distillation algorithm has a significant impact on the quality of the generated images. The detailed ablation ablation study of skipping-step technique and one-stage guided consistency distillation can be found in Section 5.2. We briefly summarize the impact of skipping-step technique and one-stage guided consistency distillation below:
>
> **Skipping-Step Technique**: This method notably enhances convergence speed, a crucial aspect of efficiency. As quantitatively demonstrated in Figure 3 (DDIM sampler), the model's convergence speed with k=1 is significantly slower compared to other settings from k=5 to k=20. In specific, it shows that at 4000 training steps, the model with k=1 has a slower convergence, evidenced by a FID of 26.8, compared to k=20 with a significantly better FID of 13.3. Moreover, at k=20, the model reaches a FID of 14.4 in just 2000 steps, matching k=1 performance at 12000 steps, achieving at least **$6\times$** convergence speed, strongly demonstrating the effectiveness of the skipping-step method.
>
> **One-stage guided consistency distillation**: Incorporating Classifier-Free Guidance (CFG) into the LCM significantly enhances image quality. Figures 4 and 5 illustrate that a larger CFG scale produces more realistic and higher-quality images. However, a very small CFG scale (which is equivalent to not using guided distillation), particularly at 2, results in lower quality generation. This enhancement in quality is validated by the CLIP scores in Figure 4 and the visibly superior image quality showcased in Figure 5.
>
> > 2. What is the computational complexity of solving augmented PF-ODE?
>
> Thanks. We would like to clarify that we first view the guided reverse diffusion process as solving an augmented probability flow ODE (Augmented PF-ODE), then  LCM is designed to effectively to solve this PF-ODE during inference stage. The computational complexity of solving this Augmented PF-ODE is essentially \textbf{equivalent} to (Number of inference steps, also named the number of function evaluation (NFE)) $\times$ (Computational cost for each step). Table 1 and 2 show the superiority of LCM on solving the augmented PF-ODE in a few inference steps (1$\sim$8). Additionally, since the classifier-free guidance (CFG) has already been distilled into LCM, we do not need to calculate the unconditional term $\epsilon_\theta(z_t,t,\varnothing)$ during each inference steps, which further reduce the computational cost in each step, resulting in significantly faster speeds and reduced GPU memory usage during inference. **We also include the inference time in **Appendix. H (Figure.7)** for your reference**. We achieve at least **$4\times$** acceleration compared with DPM-Solver++.
>
> > 3. How much faster is the inference time exactly compared with other methods?
>
> Thanks for your suggestion. We report the inference speed/FID performance in **Appendix. H (Figure.7)**. It can be seen that we achieve at least **4$\times$ acceleration** compared with DPM-Solver++.
>
> > 4.  LCMs applied to other LDMs besides Stable Diffusion?
>
> Thanks for raising this point. We would like to emphasize that Stable Diffusion, recognized as one of the most advanced and extensively used LDMs in the AI researchers and artistic communities, serves as a challenging benchmark. The successful application of LCM to Stable Diffusion has already demonstrated our method's effectiveness and potential. Having established its performance on such challenging task, we believe additional validation on simpler LDM is not essential. To further highlight LCM's promise and superiority, we also undertook a more challenging experiment applying LCM to the **SD-XL** [1], which has three times as many parameters and a resolution of 1024x1024, compared to the original SD model. Detailed generation results in **Appendix.L (Figure. 12)** further demonstrate LCM's potential and impressive performance.
>
> Reference:
> [1] Podel, et al. "Sdxl: Improving latent diffusion models for high-resolution image synthesis". arXiv preprint arXiv:2307.01952 (2023).

---

> ### Author Response · Authors · 2023-11-22
> **Reminder to Reviewer dEPF**
>
> Dear Reviewer,
>
> Thank you for reviewing our paper. We hope our responses have addressed your concerns. If so, we kindly ask if you could reconsider your score for our work.
>
> Should you have any further questions, we're ready to discuss and clarify.
>
> Many thanks for your time and effort.

---

> ### Author Response · Authors · 2023-11-22
> **Deadline coming. looking forward to your feedback.**
>
> Dear Reviewer dEPF, We kindly request your feedback as the rebuttal deadline is approaching in less than 1 days. We hope our previous feedback addresses your concern. We would like to thank you again for your time and previous review and we are looking forward to further discussions.

---

### Author Response · Authors · 2023-11-19
**General Response**

## 1. Results Update:

We thank all reviewers for the insightful comments and helpful suggestions. We have revised the paper accordingly, with revised parts marked in **purple**.
- Main Paper: We reorganized our paper, corrected typos, and added some details.
- Main Paper: We included more results for **latent consistency finetuning** (LCF) on 2 different datasets. (Figure 6)
- Appendix H: We added a comparison of actual **inference time** and **performance** metric. (Figure 7). Showing **$4\times$** acceleration compared with DPM-Solver++.
- Appendix J: We added more 2-step inference results of Dreamshaper-v7 model. (Figure 10)
- Appendix K: We added qualitative results for various ODE solvers and skipping steps. (Figure 11)
- Appendix L: We supplemented the results of distillation based on the **SDXL**[1] model, showing  more photo realistic generation results. (Figure 12)
- Appendix M: We included the prompts used for Figure 1.


## 2. About Contribution and Novelty:

We thanks the constructive feedbacks from the reviewers. And we **politely ask for a reevaluation for the LCM paper's contribution**. We want to emphasize the following points:

1) We understand the concerns raised about the perceived straightforwardness of our contribution. However, **simplicity and effectiveness, far from being limitations**, are vital principles in AI research. Why opt for complicated, fancy, and perplexing methods if a simple and efficient approach can solve the problem? Our work not only **identifies critical components for effective LDM distillation** but also **achieves impressive few-step inference results**, paving the way for real-time generative AI applications.

2) While some perceive our method as a straightforward extension of consistency distillation in latent space, it actually encompasses key differences. **First:** A key difference between the original consistency models [2] and LCM lies in the transformation of the **empirical PF-ODE (Eq.24)** from the original consistency model to a **more general PF-ODE** (Eq.8). We also adapted the Eulder/Heun solver under EDM schedule [3] to a more compatible solver of Stable Diffusion (see Eq.18 and Appendix E, including DDIM, DPM, DPM++ solvers). **Second**: Our approach extends the original consistency models's parameterization $x$-prediction (Eq.24) to a more general parameterization $\epsilon$-prediction and $v$-prediction that is more compatible for Stable Diffusion (see Eq.9 and Appendix D.). These transformation and reformulation are **non-trivial and are critical** for LCM's success. **However, we have weakened these differences in the main paper for readers much easier to read.**

3) By combining these **simple and efficient** methods, including latent consistency distillation (LCD), one-stage guided consistency distillation (CFG), skipping-step techniques, LCM is the first method that achieve **single-stage** guided distillation on high resolution text-to-image generation, demonstrating **impressive performance on few-step inference** and **remarkable efficiency**, compared with previous two-stage guided distillation method (e.g Guided-Distill [4]).


In conclusion, we argue that the conceptually simplicity of our contributions should not be seen as a lack of novelty, but rather a virtue. And **we firmly believe that LCM has made significant contribution to this field and we sincerely requires a reevaluation of the LCM contribution.**

References:

[1] Podell, Dustin, et al. ”Sdxl: Improving latent diffusion models for high-resolution image synthesis.” arXiv preprint arXiv:2307.01952 (2023).

[2] Song, Yang, et al. "Consistency models." (2023).

[3] Karras, et al. "Elucidating the design space of diffusion-based generative models" (2022)

[4] Meng, et al. "On distillation of guided diffusion models" (2023)

---

### Meta-Review · Area_Chair_5eQU · 2023-12-09

**Metareview:**

This paper proposes a latent extension of consistency models for accelerated sampling from diffusion models. While the reviewers recognize the importance of the problem, they raised concerns regarding the novelty of the proposed approach and missing ablations. Unfortunately, the ratings are mixed around the borderline, and the paper does not pass the acceptance bar at ICLR.

**Justification For Why Not Higher Score:**

The proposed approach is an application consistency models in latent space.

**Justification For Why Not Lower Score:**

N/A

---

### Decision · Program_Chairs · 2024-01-16

Reject